# Application of biological and fisheries attributes to assess the vulnerability and resilience of tropical marine fish species

**Kolliyil S. Mohamed**[©], **Thayyil Valappil Sathianandan**[ID]*[©], **Elayaperumal Vivekanandan**[©], **Somy Kuriakose**[©], **U. Ganga**[‡], **Saraswathy Lakshmi Pillai**[ID][‡], **Rekha J. Nair**[ID][‡]

ICAR—Central Marine Fisheries Research Institute (CMFRI), Kochi, Kerala, India

© These authors contributed equally to this work.
‡ These authors also contributed equally to this work.
* tvsedpl@gmail.com

## Abstract

Taking advantage of published data on life-history traits and short-term information on fishery parameters from 3132 records for 644 fish stocks along the coast of India, we calculated resilience (R) and vulnerability (V). Further, we developed an Index of Resilience and Vulnerability (IRV) for 133 species of tropical finfishes, crustaceans, and molluscs. Using 7 resilience and 6 vulnerability attributes, two-dimensional scatter plots of the resilience and vulnerability scores were generated and the Euclidean distance and angle from the origin to each point were calculated to determine IRV and the effect of fishing on fish species. By ranking the species, the top 10 highly resilient, highly vulnerable, and high-risk species (low IRV) were identified. While small-sized species with fast growth rate and low trophic level were among the highly resilient species, large predatory species such as sharks and barracudas were among the highly vulnerable and high-risk species. More than 100 of the 133 species were resilient-yet-vulnerable, and most crustaceans showed high resilience. Differences in IRV scores among species within the same family were discernible, indicating the differences in the biological characteristics and response to fishing. Sensitivity analysis indicated that an abridged IRV with 6 attributes works similar to 13 attributes and can be used in data-deficient situations. Comparison of R and V of IRV with other assessments showed different results because of divergences in the objectives, number and types of attributes, and thresholds used. These assessments do not convey the same information and therefore great care must be taken for reproducing these frameworks to other fisheries. The results of IRV analysis can be useful for stock assessments and in developing effective management measures in combination with other complementary information.

## Introduction

Marine fish resources consist of numerous stocks belonging to hundreds of species with diverse life-history traits, distributed over large geographical areas and removed by a variety of fishing methods. Tropical fisheries are more complex than their sub-tropical and temperate

**Data Availability Statement:** Mohamed Kolliyil, Sathianandan Thayyil, Vivekanandan Elayaperumal, Kuriakose Somy, Ganga Upendra, Pillai Lakshmi,

Nair Rekha (2021). Indian Marine Fish Life Histories (INMARLH) database for determining resilience and vulnerability of tropical marine species. SEANOE. https://doi.org/10.17882/82124.

**Funding:** The author(s) received no specific funding for this work.

**Competing interests:** The authors have declared that no competing interests exist.

counterparts, with the presence of more species, different life-history traits and multiple fisheries [1]. Consequently, assessing the impact of tropical fisheries is more challenging, which is compounded by the insufficient availability of operational quantitative assessment for most stocks. In this situation, the status of fish stocks is not properly understood due to deficiency in data, leading to uncertainties in identifying the right type of management measures. Fishing is considered to have caused a general decline in global fish biomass and placed many marine species under serious conservation concern [2–4]. Many of these global assessments lack sufficient representative samples from tropical fisheries and data-deficient fisheries settings [5,6].

On a global scale, semi-quantitative assessment frameworks have been developed to rapidly evaluate the risks of fishing to over 1000 marine fish populations and prioritize management and research [7]. Most of these assessments examine the impact of fishing by considering the vulnerability of a species to fishing as determined by (i) productivity–the life history characteristics which determine the intrinsic rate of population increase, and (ii) susceptibility–the interactions between population and fishing dynamics that affect the impact of the fishery on the stock [8,9]. Using some productivity and susceptibility attributes of a stock, and by assigning scores for each attribute, the productivity and susceptibility of each stock were calculated. The rank of each stock/species based on these two characteristics determined its relative capacity to sustain fishing, and therefore its priority for research and management.

In the last two decades, several modified versions of the productivity susceptibility analysis (PSA) have emerged. Depending upon the objectives of the analysis and the number and type of attributes, methodologies have been modified for better application of results. For example, after the development of risk assessment methodology for the first time to determine the sustainability of trawling for bycatch species in the Australian Northern Prawn Fishery [8], the PSA was developed as part of a hierarchical ecological risk assessment framework [9,10]. The PSA was later modified and applied to six US fisheries which were considered data-poor stocks [11]. Several other researchers also followed PSA to address ecological or fishing impacts on fish stocks and fisheries. For example, life-history traits and vulnerability assessments were used to understand the sustainability of elasmobranch fishery [12]; potential for application to fisheries management[13]; prioritizing issues for fisheries management [14]; intrinsic vulnerability of marine fish taxa in different habitats [15]; vulnerability of species in marine reserve [16]; cumulative impact of multiple fisheries on fish stocks [5]; and vulnerability of tuna longline fishery in the eastern Pacific Ocean [17].

Like other tropical fisheries, marine fisheries in India are complex and supported by multiple fisheries. Three sub-sectors in the fisheries, namely, larger mechanised boats with inboard engines (42,985 boats), smaller boats with outboard motors (66,250 boats) and non-motorised boats (25,689 boats) operate a wide variety of gears such as trawls, gillnets, lines and seines and their variants within the Indian EEZ [18], and in recent years, multiple gears i.e., trawl, gillnet and lines are operated in a single voyage depending on available species. The catches in most of these fisheries consist of a multitude of species. For example, 657 species belonging to 321 genera and 154 families are landed along the 160 km coastal stretch of the Gulf of Mannar (southeast coast of India) [19]. Collection of data on catch, effort and biological characteristics at the species level and quantitative assessment of important stocks are carried out regularly for major fisheries on a regional and national scale by the Central Marine Fisheries Research Institute. However, publications of the results of the assessments are not regular. Reference points derived either from analytical stock assessments or using empirical approaches are not available for the majority of stocks. Hence, tracking the status of stocks from quantitative assessments is not possible on a long-term basis. Regular stock assessment and reference points are required for monitoring and for determining whether the stocks are subject to overfishing or overfished and develop fishery management plans. A recent [20] national-level stock

assessment indicated that India's mean $B/B_{MSY}$ (ratio of current biomass to biomass at sustainable level) was 0.86 which is a strong reason for strengthening fisheries management. Reiterating the importance of proper assessment and monitoring [6] found that regions without assessments have little fisheries management, and stocks are in poor shape. Increased application of area-appropriate fisheries science recommendations and management tools are needed for sustaining fisheries.

One of the challenges of stock assessment and generating reference points in the tropics is that many species (small pelagics, small demersals, crustaceans and molluscs) are short-lived and their populations are renewed every year or at short intervals, making it difficult to estimate their age and growth [21]. The dynamics of these stocks are subject to high variability in recruitment, which fluctuates interannually due to fishing and environmental factors. Therefore, reference indicators such as $B_{MSY}$ (stock size that can produce maximum sustainable yield when it is fished at a level equal to $F_{MSY}$) or $F_{MSY}$ (fishing mortality that, if applied constantly year after year, would result in MSY) are generally not the most accurate methods to evaluate the status of many stocks due to specific characteristics of tropical stocks.

The absence of long-term population trend information and other limitations indicated above severely restrict the application of quantitative criteria needed for understanding the status of fish stocks. However, despite challenges in evolving stock status reference points and arriving at proper management decisions, fisheries can be well managed using precautionary management measures by a rigorous assessment of vulnerability and resilience of the species [22]. Taking advantage of the availability of data on biological characteristics, short-term information on exploitation status and population parameters for 98 species of finfish, crustaceans and molluscs from publications in peer-reviewed journals and grey literature [23], developed a Sustainability Index (siFISH) for marine fish species in India. In addition to the type of attributes used, the siFISH analysis differed from PSA by using different threshold (cutoff) values for scoring the attributes. Many PSAs used threshold values suggested by Musick [24] for calculating productivity index estimates. For example, for the von Bertalanffy growth coefficient (K), >0.3, 0.16–0.3, 0.05–0.15 and <0.05 were used as high, medium, low and very low productivity values respectively. In siFISH analysis, the K of 140 marine fish stocks belonging to 98 species in India ranged from 0.12 to > 3.0 with 97.7% of the values being above 0.3. Thus, the application of threshold values suggested by Musick [24] on fish stocks in India would result in the classification of 97.7% of stocks as highly productive in the context of K. Similar disparities in threshold values were observed for a few other attributes related to life-history traits such as fecundity, L∞ etc. The threshold values, based on biological reference points, were suggested by Musick [24] to define the extinction risk of marine fishes (temperate species such as Sandbar shark, Blue grenadier, Cape hake, Bay anchovy, Orange roughy etc) with a rider that the values may not be consistent within all productivity estimates because of the great diversity in life-history strategies among fishes. Realising this, different threshold values appropriate for tropical stocks were identified and used for siFISH analysis [23].

Recognizing the importance of vulnerability assessments to provide a framework for evaluating fishing impacts over a broad range of species with the available information, we considered the data used for siFISH assessment, but effected three changes in the analysis: (i) species coverage was increased to 133 including the 98 species covered for siFISH assessment; (ii) data coverage was extended from the period 1985–2008 to 1954–2015, and (iii) method of analysis was revised to calculate resilience (R) and vulnerability (V) and further to develop an Index of Resilience and Vulnerability (IRV). Resilience here is defined in terms of the ability of a system (species) to absorb shocks, to avoid crossing a threshold into an alternate and possibly irreversible new state, and to regenerate after disturbance [25] and vulnerability means the extent to which changes can hurt or harm a species.

The new index has provided an opportunity to compare the results of the analysis with complementary resilience and vulnerability analyses as well as identifying the scope of application of the results to fisheries management in data-deficient tropical fisheries settings. It has also given an impetus for a discussion on further steps to take forward the analysis.

## Materials and methods

### Data source

Biological and fisheries information on marine fish and shellfish species published in research journals on the Indian fisheries in the Arabian Sea and Bay of Bengal were collated into a database called INMARLH (Indian Marine Fish Life Histories). The major journals referred, namely, the Indian Journal of Fisheries (from 1954) and the Journal of the Marine Biological Association of India (from 1959) and the number of papers used for obtaining information from all sources are shown in **S1 Table**. The annual reports of Central Marine Fisheries Research Institute (CMFRI; which is the national agency responsible for stock assessments, www.cmfri.org.in) from 2002 were accessed to screen biological and fishery information from research projects executed by CMFRI. The database contained 3132 records on 644 stocks from the 9 maritime states of India (Fig 1). The 644 stocks belonged to 133 species; 90 genera; 55 families and 19 orders. This database is available from a public repository https://www.seanoe.org/ (data upload reference # 82124 –doi pending).

### Species and attributes selected

The primary criterion used to select attributes was the availability of data on different biological and fishery characteristics. Those species with the availability of information on the maximum number of attributes were selected, and thus, 13 attributes were selected. The species selected included 96 teleosts, 6 elasmobranchs, 20 crustaceans, 3 each of bivalves and gastropods and 5 cephalopods (**S2 Table**). All 133 species did not have data on all 13 attributes (**Table 1**).

The 13 attributes selected were grouped into 7 biology and 6 fisheries attributes (**Table 2**). The biological attributes were those on characteristics that inherently increase the resilience of the species and the fishery attributes were those relating to the vulnerability of the species to fishing. The terms resilience and vulnerability were considered as contraries, and consequently, those species with high resilience scores were theoretically considered less vulnerable and vice versa.

The attributes were scaled into ranks ranging from 1 (low), 2 (medium) and 3 (high) for resilience and vulnerability separately. The logic used to numerically rank the attributes is described below and also shown in Table 2.

### Resilience (R) attributes

**1. Rate of growth: Growth coefficient (K).**   This is the rate at which the fish approaches its maximum size and is a key factor in the von Bertalanffy growth function [26]. A large long-lived fish would generally have a low K value and a small or medium-sized short-lived fish would have higher K values. In general, tropical marine species have faster growth rates as compared to temperate species. The database contained 644 records of annual K values ranging from 0.12 to 3.9.

**2. Longevity: Asymptotic length (L∞).**   This is the theoretical maximum size of the fish used in the von Bertalanffy growth function and has a bearing on the longevity of the species [26]. It is derived by plotting the size-frequency polygons against time using several iterative

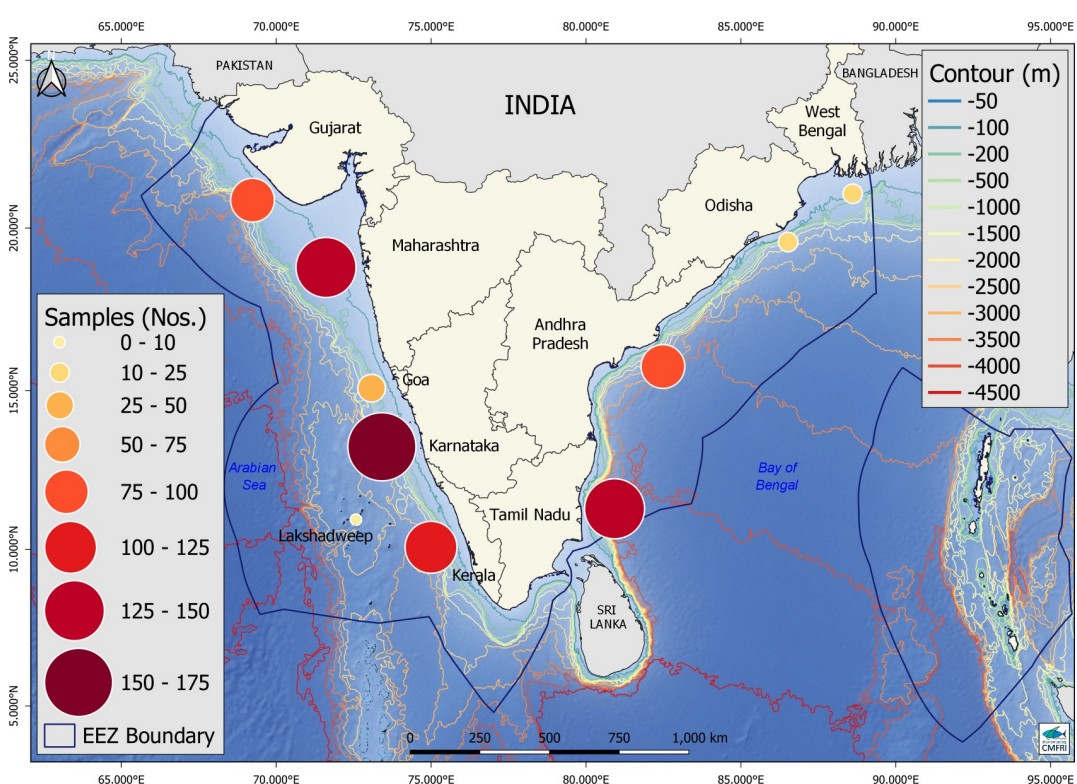

**Fig 1. Map of coastal India showing the Arabian Sea, Bay of Bengal, the maritime state boundaries, and EEZ.** Circles indicate the distribution of data records used in the study from different ecoregions. Base map and administrative boundaries sourced from the Natural Earth (http://www.naturalearthdata.com/); bathymetric data from the GEBCO gridded bathymetry data (GEBCO 2014 Grid, www.gebco.net); and EEZ boundaries from the Flanders Marine Institute Maritime Boundaries Geodatabase, version 11 (https://www.marineregions.org/).

methods. A large fish that lives longer tends to be less resilient. The INMARLH database showed that most fishes had L∞ between 200 and 400 mm. For species with no record of L∞ values (42 stocks), the maximum length recorded was used as a proxy. The database contained 644 records of L∞ values ranging from 31.1 to 3615 mm.

**Table 1. Descriptive statistics of 11 attributes (omitting ratios and price values) for 644 marine fish stocks used for deriving R and V scores.**

| Statistic | K | L∞ | Lm | NSM | Fecundity | BLD | CPI | MTL | Dist | Er | Lr |
|---|---|---|---|---|---|---|---|---|---|---|---|
| Count (n) | 644 | 644 | 593 | 528 | 605 | 634 | 644 | 644 | 644 | 611 | 577 |
| Average | 0.96 | 449.04 | 223.59 | 5.73 | 578394 | 4.24 | 360.56 | 3.45 | 66.78 | 0.61 | 123.45 |
| CV | 51.78 | 90.37 | 93.64 | 74.41 | 182 | 51.84 | 68.20 | 18.58 | 42.38 | 34.05 | 105.25 |
| Q1 | 0.61 | 199.75 | 87.00 | 3.00 | 14000 | 3.03 | 81.50 | 2.94 | 38.28 | 0.50 | 40.00 |
| Q2 | 0.85 | 314.00 | 150.00 | 4.00 | 134000 | 3.91 | 298.60 | 3.50 | 76.77 | 0.63 | 71.00 |
| Q3 | 1.30 | 522.00 | 275.00 | 7.00 | 477282 | 4.80 | 662.80 | 4.14 | 91.18 | 0.73 | 129.00 |
| Q4 | 3.90 | 3615.00 | 1750.00 | 12.00 | 4723000 | 19.10 | 662.80 | 4.50 | 100.00 | 0.93 | 1000.00 |
| Min | 0.12 | 31.10 | 14.00 | 1.00 | 1.00 | 1.20 | 16.10 | 2.00 | 5.38 | 0.13 | 6.00 |
| Max | 3.90 | 3615.00 | 1750.00 | 12.00 | 4723000 | 19.10 | 662.80 | 4.50 | 100.00 | 0.93 | 1000.00 |
| SD | 0.50 | 405.82 | 195.04 | 3.00 | 1008579 | 2.13 | 245.92 | 0.64 | 28.30 | 0.16 | 116.37 |

CV–coefficient of variation; Q1-Q4 (Quartile Q1 and Q3 are the first and third quartiles and Q2 is the median value; Q4 is is the maximum value); Min–Minimum; Max–Maximum; SD–Standard deviation. Attributes are K (growth coefficient); L∞ (Asymptotic length); Lm (Length at maturity); NSM (Number of spawning months); BLD (Body length depth); CPI (Coastal productivity index); MTL (Mean trophic level); Dist (Distribution); Er (Exploitation rate) and Lr (Length at recruitment).

**Table 2. Ranking criteria and logic [24,26] of different resilience and vulnerability attributes used in IRV.**

| Attribute | Rank | Criteria | Logic |
|---|---|---|---|
| **Resilience attributes** | | | |
| 1. Growth coefficient (K) | 1 | 0.1–0.6 | Species which exhibit high K values are more resilient. |
| | 2 | 0.61–1.2 | |
| | 3 | > 1.2 | |
| 2. Asymptotic length (L∞) | 1 | > 800 mm | Small size species which exhibit lower L∞ values are more resilient. |
| | 2 | 401–800 mm | |
| | 3 | < 400 mm | |
| 3. Length at maturity (Lm)/L∞ ratio (abs) | 1 | (Lm/L∞ -0.5) = > 0.20 | This ratio indicates the reproductive load of the species. A ratio of 0.5 (median) was ranked highest, and the least deviation from the median was ranked more resilient. |
| | 2 | (Lm/L∞ -0.5) = 0.10–0.20 | |
| | 3 | (Lm/L∞-0.5) = < 0.10 | |
| 4. Number of spawning months (NSM) | 1 | 1 to 4 | A species spawning in all months of the year is more resilient. |
| | 2 | 5 to 8 | |
| | 3 | 9 to 12 | |
| 5. Fecundity (Fc) | 1 | < 25,000 | Species with high fecundity are more resilient. |
| | 2 | > 25,000–100,000 | |
| | 3 | > 100,000 | |
| 6. Coastal productivity index (CPI) | 1 | < 150 | Estimated as a product of monthly Coastal Upwelling Index (CUI) and Chlorophyll-*a* concentrations from 30 lat-long positions along the Indian coast for the period 1998–2008. This was linked to the species distribution index to arrive at the CPI. Species maximally distributed in highly productive waters are more resilient. |
| | 2 | 151–400 | |
| | 3 | > 400 | |
| 7. Mean trophic level (MTL) | 1 | > 4.0 | Lower trophic level fishes are more resilient. |
| | 2 | 3.0–4.0 | |
| | 3 | 2.0–3.0 | |
| **Vulnerability attributes** | | | |
| 8. Species geographic Distribution (Dist) | 1 | > 65 | Species which have wider geographic distribution are less vulnerable. Species occurrence from catch database across Indian states was used to convert into the area. |
| | 2 | 35–65 | |
| | 3 | < 35 | |
| 9. Body length depth (BLD) ratio | 1 | > 15 | A high ratio indicates species that have a higher probability of escapement through nets, and therefore, are less vulnerable. |
| | 2 | 5–15 | |
| | 3 | < 5 | |
| 10. Exploitation Ratio (Er)—Ratio of fishing mortality to total mortality (F/Z) | 1 | < 0.5 | Species with low Er values are less vulnerable. |
| | 2 | 0.5–0.7 | |
| | 3 | > 0.7 | |
| 11. Length at recruitment (Lr)/L∞ ratio | 1 | > 0.5 | Species which recruit at larger sizes (higher ratio) are less vulnerable. |
| | 2 | 0.3–0.5 | |
| | 3 | < 0.3 | |
| 12. Gear susceptibility (G) | 1 | > 12.5 | The gear ecosystem effects rank of [27] and its product with weighted gear-wise catch is used. High values indicate less vulnerability. |
| | 2 | 7.5–12.5 | |
| | 3 | < 7.5 | |
| 13. Landing Price of species (P) | 1 | High | Species with low market price was given a higher ranking (less vulnerability), considering that it is not targeted by fishers. |
| | 2 | Medium | |
| | 3 | Low | |

All attributes have 3 ranks. Higher rank (#3) indicates high resilience and low vulnerability. Criteria are based on the broad distribution of values (min-max).

**3. Reproductive load.** Fish usually reproduce when they have reached about half of the maximum size they are likely to reach (Lmax or L∞). The size at which 50% of the population is mature is called the size at first maturity (Lm), and the fraction Lm/L∞, called the reproductive load, tends to be higher in small than in large fish [26]. A ratio of 0.5 (median) was ranked highest, and other ranks were based on the deviation from the median. The database contained 593 records of reproductive load values ranging from 0.1 to 0.8.

**4. Spawning frequency: Number of spawning months (NSM).** This rank was based on the number of months a species spawns in a year. A species spawning throughout the year (12 months) was ranked the highest and considered more resilient. The database contained 528 records of NSM values ranging from 1 to 12.

**5. Number of eggs produced: Fecundity.** Fecundity (defined as the number of eggs ripening between the current and next spawning period in a female) measurements are usually *in-situ* observations and do not indicate annual reproductive potentials. A species having high fecundity was ranked the highest and considered as more resilient. The database contained 605 records of fecundity values ranging from 1 to 4.7 million eggs.

**6. Habitat productivity: Coastal productivity index (CPI).** The productivity of the habitat in which a species is distributed is considered to have an impact on the overall production of the stock. CPI was estimated as a product of monthly Coastal Upwelling Index (CUI) and Chlorophyll-*a* concentration from 30 latitude-longitude positions from the 9 maritime states of the Indian coast for the period 1998–2008. The monthly values were averaged to arrive at annual values for each maritime state. This was linked to the species geographic distribution to arrive at the CPI. Species having wide distribution in highly productive waters were given higher scores. The database contained 644 records of CPI scores ranging from 16.1 to 662.8.

**7. Position in the food web: Mean trophic level (MTL).** The trophic level of an organism is the position it occupies in a food chain or trophic pyramid. The mean trophic levels were estimated from diet studies, from developed trophic models [28] and FishBase [26]. The highest rank was given to species with low trophic level as these species were considered resilient. The database contained 644 records of MTL values ranging from 2.0 to 4.5.

## Vulnerability (V) attributes

**8. Species geographic distribution.** A species with wide cosmopolitan distribution is presumed to be adaptive and less vulnerable. The catch record in each maritime state was taken as a surrogate for the distribution of species. The cumulative area of maritime states contributing to the catch was considered for arriving at the area of distribution of species in $km^2$. Species distributed in larger areas were given a higher rank. The database contained 644 records of geographic distribution scores ranging from 5.38 to 100, and the range was equally divided for assigning the 3 ranks.

**9. Probability of escapement from gear: Body length depth ratio (BLD).** This ratio determines the possibility of escape of the fish through fishing nets. The highest rank was given to species with a high ratio considering the higher probability of escapement of the species from fishing nets. Fishes such as eels and ribbonfishes have a high ratio, and therefore are less vulnerable to fishing. The database contained 634 records of BLD ratios ranging from 1.2 to 19.1.

**10. Exploitation ratio (Er = Fishing mortality/Total mortality).** The exploitation ratio of a stock is the proportion of the numbers or biomass removed by fishing [26]. Er was a calculated ratio of fishing mortality to total mortality (F/Z). A species with a low ratio was given a lower ranking and is, therefore, less vulnerable to fishing pressure. The database contained 611 records of E ratios ranging from 0.1 to 0.9.

**11. Recruitment.** Recruitment is the process in which young fish enter the fishery by being caught with the fishing gear. The ratio of length at recruitment (Lr) by the maximum or asymptotic length indicates the relative vulnerability of a species to capture [26]. Species which recruit at larger sizes are less vulnerable and given a higher ranking. The database contained 577 records of Lr/L∞ ratios ranging from 0.02 to 0.8.

**12. Susceptibility to gear.** Catch susceptibility score was assigned for each species to different gears following [27] and a product of these values with a proportional weightage of catch contribution by different gears to each species was obtained which was scaled to the rank. A species with a high rank is considered to be less susceptible to fishing gear, and therefore, less vulnerable. The database contained 634 records of gear scores ranging from 1.2 to 19.1.

**13. Price information.** Highly-priced species in the catch are always a target for the fisheries and given a score of 1. Species with a low market price was given the higher ranking of 3 (less vulnerability), considering that it is not targeted by fishers.

## Scoring of attributes

The database was created in MS Access. Based on the ranking criteria given in Table 2, the ranks (between 1 and 3) were assigned to each attribute for all 644 data records of the 133 species and averaged for species.

## Plotting the scores

Two-dimensional scatter plot of the resilience and vulnerability scores were generated with resilience scores as abscissa (in reverse order) and vulnerability scores as ordinates. Plots were created separately for teleosts (two plots), elasmobranchs, crustaceans, and molluscs. A similar plot was created to show how taxonomic families are distributed.

## Determining the index of resilience and vulnerability (IRV) score

There are two aspects in the resilience-vulnerability plot that determine the capacity of a species to sustain fishing pressure. The first one is the Euclidian distance from the origin to the point in the graph corresponding to the species and the second one is the angle this line makes with the x-axis (resilience). When both the distance and angle reduce, resilience increases and vulnerability decreases. When both distance and angle increase the vulnerability increases and resilience decreases. Thus, we have used the following quantity as the IRV score which depicts the status of the species, higher IRV score (scaled from 0 to 1) means the species is in the safe zone, and conversely, a low IRV score indicates that the species is under high-risk of depletion. The coordinates of the origin in the graph is (3,1) and the x and y coordinates of any point in the graph are restricted between 1 and 3. If $(x_i, y_i)$ is the coordinates of a point in the graph, then the Euclidian distance from the origin and the angle of the line with x-axis can be calculated as,

$$d_i = \sqrt{(3 - x_i)^2 + (y_i - 1)^2}$$

$$\theta_i = \tan^{-1}\left(\frac{y_i}{x_i}\right)$$

The maximum possible distance for a point in the graph is $\sqrt{8}$ and the maximum angle is $\frac{\pi}{2}$ radians. Hence, we defined IRV score ranging from 0 to 1 as:

$$IRV = \left(1 - \frac{d_i}{\sqrt{8}}\right)\left(1 - \frac{\theta_i}{(\pi/2)}\right)$$

The IRV scores of the 6 major groups (teleosts, elasmobranchs, crustaceans, bivalves, cephalopods and gastropods) were subject to a one-way ANOVA and then pairwise comparisons were made using the Student t-test.

## Sensitivity analysis

To identify the R and V attributes that are the most influential in determining the final scores, a sensitivity analysis was carried out using Alexander's S which measures the sensitivity of an outcome to changes in a selected input variable [29]. The sensitivity analysis is also aimed at deriving an abridged set of R and V attributes which can relatively accurately determine the R and V scores in data-limited situations. The expression for Alexander's S is:

$$S = \frac{\sum_{i=1}^{N} \frac{(O_{ik} - O_{ij})^2}{O_{ij}}}{max\left(\sum_{i=1}^{N} \frac{(O_{ik} - O_{ij})^2}{O_{ij}}\right)}$$

Where, $O_{ij}$ is the previous value of the outcome and $O_{ik}$ is the current value of the outcome. For each of the attributes, the values were sorted and changes in ranks total considering all the attributes average ranks were taken as the output to determine the sensitivity of the attributes. The 6 most sensitive attributes were again used to create an abridged version of the R-V plot for the 10 most resilient and vulnerable species, and the difference in Euclidean distance was calculated.

## Comparison of IRV with other vulnerability assessments

The IRV analysis was compared with the following three other vulnerability assessments:

i. A select subset of 11 species was subjected to the standard Productivity Susceptibility Analysis (PSA). PSA was originally developed by [8]. The PSA is a semi-quantitative and rapid risk assessment tool that relies on the life history characteristics of a stock (i.e., productivity) and its susceptibility to the fishery in question and has been widely used [11,30,31]. For running the PSA, the Excel Macro used by the Marine Stewardship Council was used (*https://www.msc.org › scheme-documents › forms-and-templates*). The PSA scores were compared with IRV scores to determine the differences.

ii. [32] created a method that employs a fuzzy logic expert system to determine species intrinsically vulnerable in the context of fishing pressure calculated based on the species life history and ecological characteristics. This methodology is used in vulnerability assessment in FishBase [26], the largest and the most extensive fish data information system, for almost all fish species including the 102 species (teleosts and elasmobranchs) tested for IRV analysis. In the analysis, the species are designated into 4 vulnerability categories from 'low' to 'very high'. The intrinsic vulnerability was also expressed on an arbitrary scale from 1 to 100, with 100 being the most vulnerable. FishBase has also calculated the resilience of each species and identified the species into 4 categories from 'very low' to 'high' resilience following [24]. By segregating the vulnerability and resilience score of IRV analysis ranging from 0 to 3 into four equal categories, the species scores were compared with the 4 categories of FishBase.

iii. The Red List prepared by the International Union for Conservation of Nature (IUCN) has classified the species into 9 categories namely, Not Evaluated, Data Deficient, Least Concern, Near Threatened, Vulnerable, Endangered, Critically Endangered, Extinct in the Wild and Extinct. The status of the 133 species in those listed by IUCN was compared with the vulnerability score calculated in the IRV analysis.

# Results

## Distribution of ranks

The descriptive statistics for attributes are shown in Table 1. When the attributes were ranked as per Table 2, the higher frequency for median rank (2) was observed for 7 of the 13 attributes —K, NSM, MTL, CPI, ER, Gear and Price which means that it followed a normal distribution. A high frequency was observed for higher rank (3) for 5 attributes (L∞, Lm, FC, BLD and LR) and low rank with high frequency was observed for only one attribute (Distribution). In general, most fish stocks had small body size, had higher fecundity and lower reproductive loads, lower BLD ratios and recruited into the fishery at smaller sizes. The majority of the fish stocks also had wide geographic distribution.

## Resilience and vulnerability scores

The calculated resilience (R) scores showed a normal distribution pattern which was slightly skewed towards higher resilience (Fig 2A). More than 100 of the 133 species had relatively high resilience scores between 2 and 2.5. Very high resilience (>2.5) was observed for 18 species which included crabs, shrimps, whelk and fishes. The top-10 highly resilient species (Table 3) included 5 crustaceans, 4 teleosts and 1 gastropod.

The estimated vulnerability (V) scores also showed a normal distribution pattern which was skewed towards high vulnerability (Fig 2B). Moderately high V scores of 2.5 were observed for 80 species and a median score of 2 was observed in 41 species. There were 11 species that were highly vulnerable (>2.5), and these did not include any crustaceans and included some gastropods, sharks and teleosts. The top-10 highly vulnerable species (Table 3) included 6 teleosts, 2 elasmobranchs and 2 gastropods. Based on the IRV scores, the species with low scores were high-risk or most risky species (Table 3). The top-10 in this list included 3 sharks and 1 ray, 5 teleosts (great barracuda, flat needlefish, bronze croaker, streaked seerfish, smallhead hairtail) and the gastropod, sacred chank.

More than 75% of the species were medially resilient and vulnerable (Table 4). None of the elasmobranchs examined showed high resilience. The Indian Babylon (*Babylonia zeylanica*) and the tooth pony fish (*Gazza minuta*) showed both high resilience and high vulnerability. High vulnerability and low resilience were observed in the flat needlefish (*Ablennes hians*), the spot-tail shark (*Carcharhinus sorrah*) and the great barracuda (*Sphyraena barracuda*). Seventeen species showed high vulnerability. Among teleosts, about 90% of the species were medially resilient and vulnerable (Table 4). All the elasmobranchs studied were highly or medially vulnerable with medium or low resilience. Most crustaceans were highly or medially resilient with medium vulnerability. In molluscs, the sacred chank, *Turbinella pyrum* was highly vulnerable and all squids and cuttlefishes were medially resilient and vulnerable.

## Resilience and vulnerability plots and IRV scores

The composite IRV plot for 133 species is given in Fig 3. The points are clustered between 1 and 3 for resilience and between 1.5 and 3 for vulnerability. To unclutter the plot, separate plots were made for teleosts (1 to 44 and 45 to 96: numbers as per S2 Table); elasmobranchs

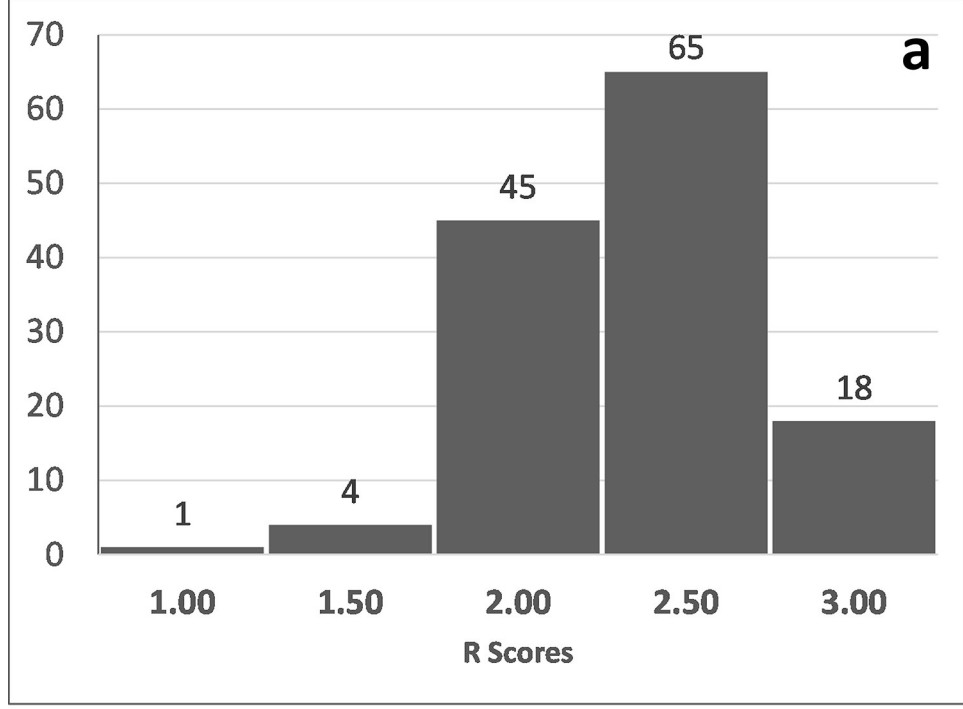

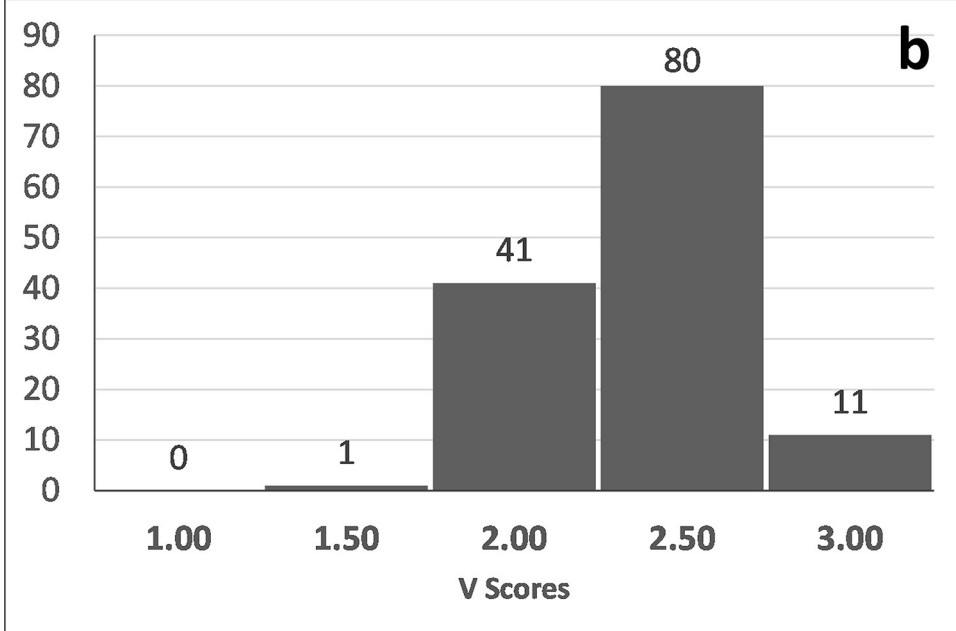

**Fig 2.** Frequency distribution of calculated resilience (R) scores for 133 species/stocks (a) **and** frequency distribution of calculated vulnerability (V) scores for 133 species/stocks (b).

(97 to 102); crustaceans (103 to 122) and molluscs (123 to 133). The white sardine, *Escualosa thoracata* (#27; Fig 4A) had high resilience and relatively low vulnerability, while at the other end of the spectrum, the flat needlefish, *Ablennes hians* had a very high vulnerability and low resilience. The majority of the teleost species had medium resilience and median vulnerability (Fig 4A and 4B). All fishes in the family Ariidae (catfishes) had medium resilience and medium

**Table 3. Top 10 resilient and vulnerable species based on R and V scores; species code is provided to locate the position of the species in the IRV plot; species that scored low IRV have been designated as 'risky'.**

| No. | Most Resilient Species | Species Code | R Score | Most Vulnerable Species | Species Code | V Score | Most risky species | Species Code | IRV |
|---|---|---|---|---|---|---|---|---|---|
| 1 | *Portunus sanguinolentus* | 120 | 2.873 | *Turbinella pyrum* | 133 | 2.833 | *Carcharhinus sorrah* | 98 | 0.022 |
| 2 | *Sardinella fimbriata* | 31 | 2.833 | *Scomberomorus lineolatus* | 84 | 2.800 | *Ablennes hians* | 9 | 0.046 |
| 3 | *Solenocera choprai* | 121 | 2.833 | *Johnius borneensis* | 63 | 2.733 | *Sphyraena barracuda* | 88 | 0.049 |
| 4 | *Charybdis feriatus* | 104 | 2.8 | *Babylonia spirata* | 131 | 2.667 | *Rhizoprionodon acutus* | 100 | 0.077 |
| 5 | *Escualosa thoracata* | 27 | 2.8 | *Sphyraena barracuda* | 88 | 2.667 | *Turbinella pyrum* | 133 | 0.081 |
| 6 | *Pseudorhombus arsius* | 10 | 2.8 | *Upeneus taeniopterus* | 55 | 2.667 | *Scomberomorus lineolatus* | 84 | 0.092 |
| 7 | *Parapenaeopsis hardwickii* | 113 | 2.75 | *Rhizoprionodon acutus* | 100 | 2.625 | *Sphyrna lewini* | 102 | 0.109 |
| 8 | *Solenocera crassicornis* | 122 | 2.75 | *Ablennes hians* | 9 | 2.600 | *Eupleurogrammus muticus* | 94 | 0.110 |
| 9 | *Babylonia zeylanica* | 132 | 2.667 | *Carcharhinus sorrah* | 98 | 2.600 | *Otolithoides biauritus* | 75 | 0.127 |
| 10 | *Gazza minuta* | 47 | 2.667 | *Upeneus vittatus* | 56 | 2.583 | *Glaucostegus granulatus* | 99 | 0.127 |

to high vulnerability (Fig 4A). The false trevally, *Lactarius lactarius*, had both high resilience and high vulnerability and was an outlier in the plot (#44; Fig 4A).

Fishes of the family Leiognathidae, which are ubiquitous in Indian marine ecosystems were found to have a low vulnerability and medium resilience (#45 to 52; Fig 4B). All scombrids, except the streaked seerfish had medium resilience and vulnerability. The streaked seerfish

**Table 4. Contingency matrix of R and V scores for all species, teleosts, elasmobranchs, crustaceans and molluscs based on Low—<1.5; Medium—> = 1.5 & <2.5 and High >2.5 scores.**

| | | Resilience | | | |
|---|---|---|---|---|---|
| **Vulnerability** | **All Species** | **High** | **Medium** | **Low** | **Total** |
| | High | 2 | 12 | 3 | 17 |
| | Medium | 23 | 92 | 1 | 116 |
| | Low | 0 | 0 | 0 | 0 |
| | **Total** | **25** | **104** | **4** | **133** |
| **Vulnerability** | **Teleosts** | | | | |
| | High | 1 | 6 | 2 | 9 |
| | Medium | 13 | 73 | 1 | 87 |
| | Low | 0 | 0 | 0 | 0 |
| | **Total** | **14** | **79** | **3** | **96** |
| **Vulnerability** | **Elasmobranchs** | | | | |
| | High | 0 | 2 | 1 | 3 |
| | Medium | 0 | 3 | 0 | 3 |
| | Low | 0 | 0 | 0 | 0 |
| | **Total** | **0** | **5** | **1** | **6** |
| **Vulnerability** | **Crustaceans** | | | | |
| | High | 0 | 1 | 0 | 1 |
| | Medium | 10 | 9 | 0 | 19 |
| | Low | 0 | 0 | 0 | 0 |
| | **Total** | **10** | **10** | **0** | **20** |
| **Vulnerability** | **Molluscs** | | | | |
| | High | 1 | 3 | 0 | 4 |
| | Medium | 0 | 7 | 0 | 7 |
| | Low | 0 | 0 | 0 | 0 |
| | **Total** | **1** | **10** | **0** | **11** |

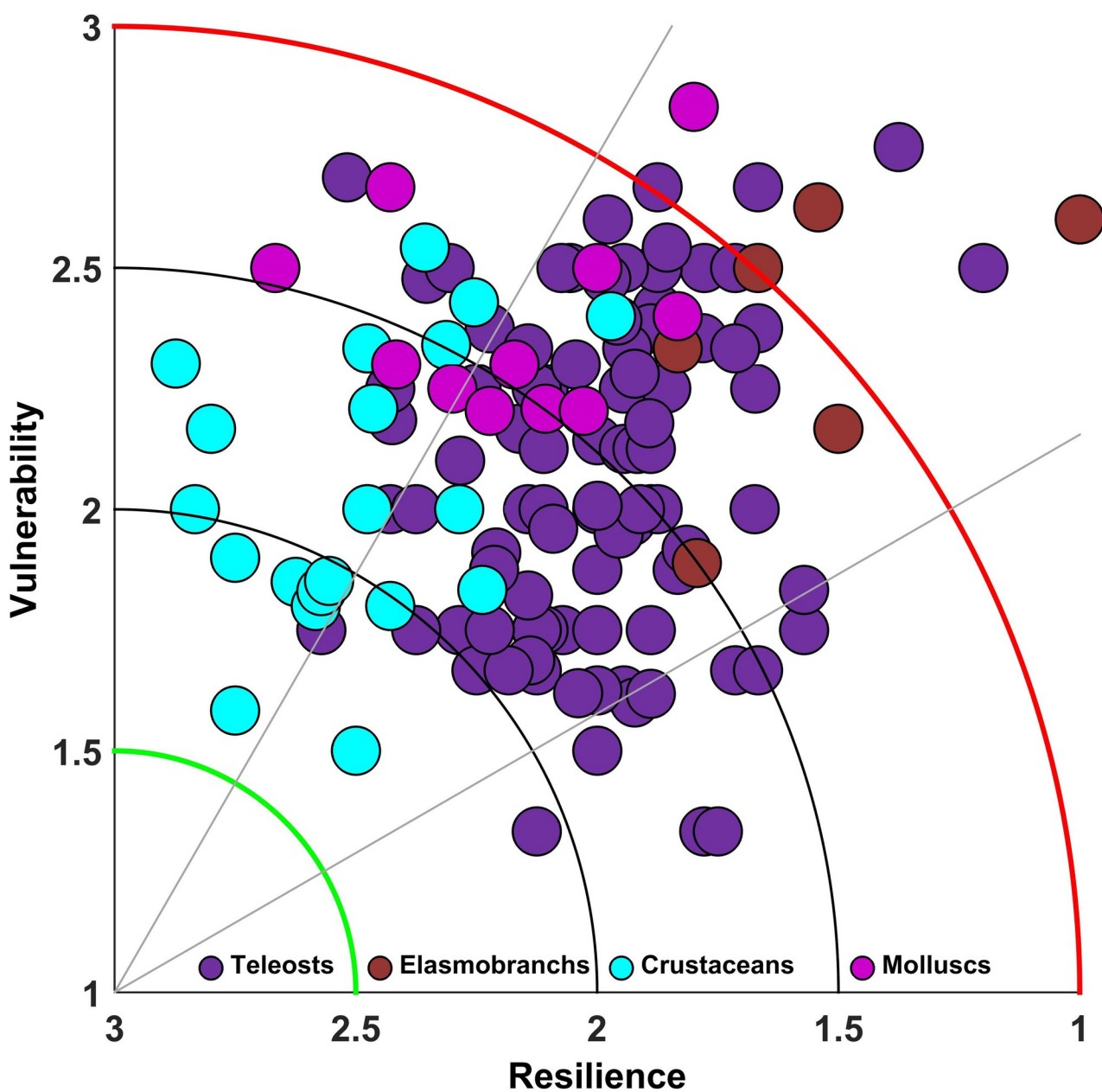

**Fig 3. Scatter plot of R-V scores for all 133 species/stocks including 96 teleosts, 6 elasmobranchs, 20 crustaceans and 11 molluscs.**

(*Scomberomorus lineolatus*) had low resilience and high vulnerability (#84, Fig 4B). All the elasmobranchs assessed had low resilience and high vulnerability (Fig 4C) and the least vulnerability was observed for the spade nose shark, *Scoliodon laticaudus* (#101, Fig 4C).

Most crustaceans had a resilience score above 2 indicating their relative hardiness (Fig 4D). Crustaceans that showed higher vulnerability were the mud spiny lobster, *Panulirus polyphagus*; penaeid shrimps, *Penaeus monodon*, *P. semisulcatus* and *Metapenaeus monoceros*. Solenocerid and palaemonid shrimps such as *Solenocera choprai*, *S. crassicornis* and *Nematopalaemon tenuipes* showed very high resilience and low vulnerability. Unlike crustaceans, molluscs show a shift in the plot towards higher vulnerability, while showing medium resilience (Fig 4E).

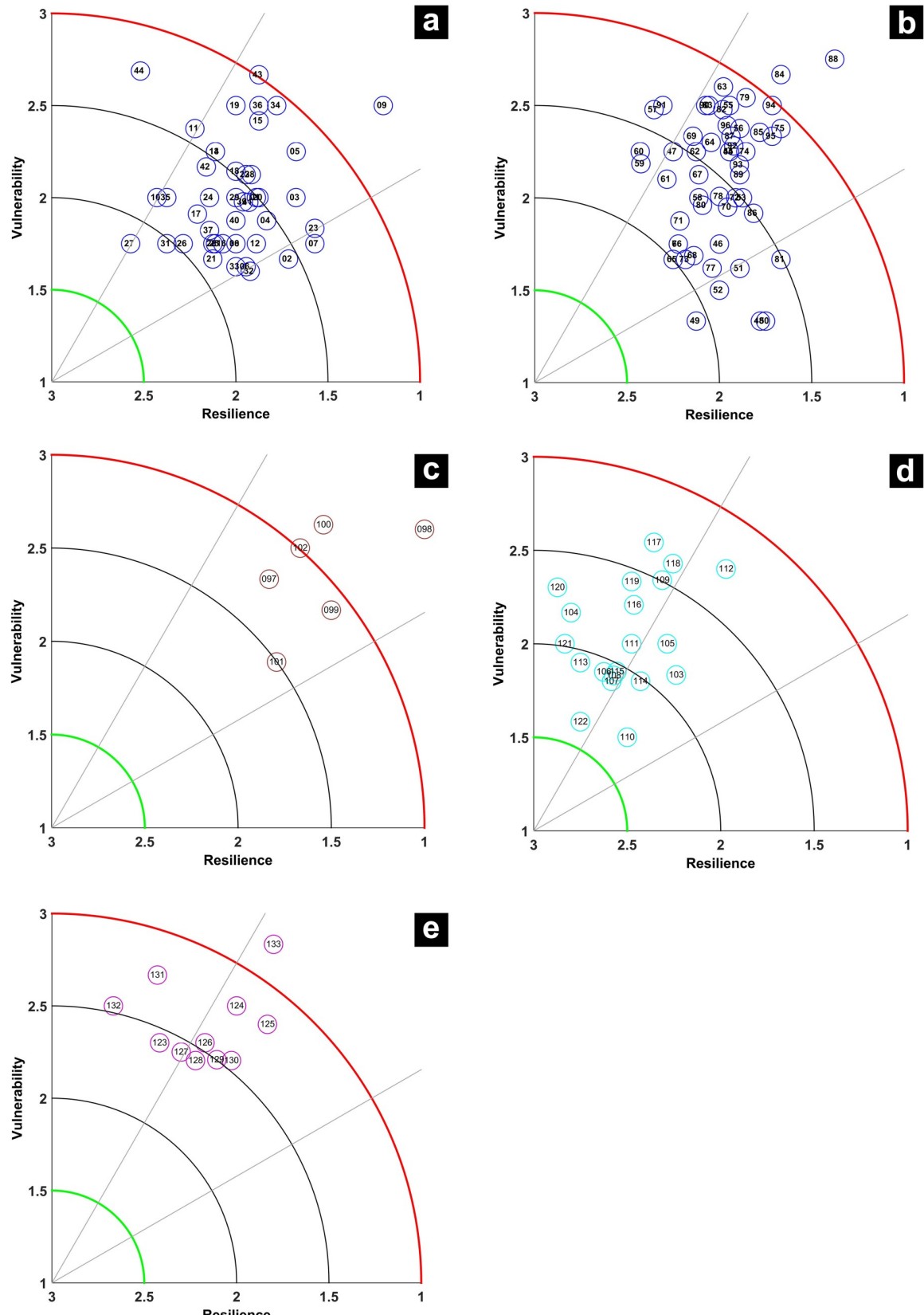

**Fig 4.** Scatter plot of R-V scores for 44 species of teleosts (a), 52 species of teleosts (b), 6 species of elasmobranchs (c), 20 species of crustaceans (d) and 11 species of molluscs (e). Species identity numbers are as per S2 Table.

**Table 5. Pairwise mean comparison of Student t-statistic of IRV scores of 6 groups of 133 species tested.**

|  | Teleosts | Elasmobranchs | Crustaceans | Bivalves | Cephalopods |
|---|---|---|---|---|---|
| **Teleosts** |  |  |  |  |  |
| **Elasmobranchs** | 3.509** |  |  |  |  |
| **Crustaceans** | 5.126** | 5.879** |  |  |  |
| **Bivalves** | 1.106 | 1.171 | 3.083** |  |  |
| **Cephalopods** | 0.238 | 2.258* | 2.739** | 0.739 |  |
| **Gastropods** | 1.537 | 0.814 | 3.490** | 0.309 | 1.084 |

** indicates high significance (p<0.001) and

* indicates significance (p<0.05). One-way ANOVA showed significant differences between groups at a 1% level.

While squids and cuttlefishes are medially resilient and vulnerable, the gastropods are mostly highly vulnerable and resilient, except for the sacred chank, *Turbinella pyrum*, which is highly vulnerable and less resilient. The bivalve window-pane oyster, *Placuna placenta*, a species that is protected under the Indian Wildlife Protection Act is both highly vulnerable and medially resilient (#124, Fig 4E).

The calculated composite IRV score showed a normal distribution pattern with 54 of the 133 species scoring a median 0.3 value. Although IRV can range from 0 (very vulnerable) to 1 (very resilient), the values we obtained ranged from a very vulnerable 0.022 (*Carcharhinus sorrah*–spot-tail shark) to resilient 0.512 (*Solenocera crassicornis*—coastal mud prawn). None of the species were close to the very resilient score of 1 (Table 6). Species that were more resilient such as the crustaceans and clupeids scored a higher index value of >0.3. The average IRV scores were 0.120 (elasmobranchs), 0.201 (molluscs), 0.237 (teleosts) and 0.340 (crustaceans) in increasing order. ANOVA showed significant (p<0.001) differences between the IRV scores of different groups. Pairwise comparisons (Table 5) showed highly significant (p<0.001) differences between elasmobranchs and teleosts and crustaceans. Crustaceans were also significantly different (p<0.001) from all molluscs. However, the level of significant difference between elasmobranchs and cephalopods was only 5%. The IRV scores and descriptive statistics by taxonomic order are shown in Table 6. The mean IRV was highest for order Decapoda and Stomatopoda and lowest for Carcharhiniformes. The SD and QD were highest for Perciformes. Differences in IRV were observed among species within the same family. IRV calculated for 7 families, each with more than 5 species showed that the coefficient of variation (CV) of IRV ranged from 18.6% (Leiognathidae) to 34.9% (Scombridae).

## Relationship between resilience and vulnerability

Vulnerability is expected to be high in species with low resilience [33]. However, the plot of vulnerability score against resilience score showed only a weak relationship with considerable scatter in the points (Fig 5). On the other hand, R and IRV were positively related with the increase in R corresponding to an increase in IRV (Fig 6A). Conversely, an increase in V led to a decrease in IRV (Fig 6B).

## Sensitivity analysis of attributes

The sensitivity of the attributes, or in other words, the most important attributes which influence the IRV scores were tested using Alexander's S method. Results indicated that the exploitation ratio (ER), growth coefficient (K), asymptotic length (L∞), coastal productivity index (CPI), geographic distribution (Dist) and the number of spawning months (NSM) were the 6

**Table 6. Distribution of IRV scores by taxonomic order in the 133 species and their descriptive statistics.**

| No. | Order | Count | Min | Max | Mean | Std Dev | Q1 | Q2 | Q3 | Q4 | CV% | QD |
|---|---|---|---|---|---|---|---|---|---|---|---|---|
| 1 | Siluriformes | 6 | 0.128 | 0.309 | 0.208 | 0.057 | 0.173 | 0.213 | 0.224 | 0.309 | 27.11 | 0.03 |
| 2 | Perciformes | 65 | 0.038 | 0.430 | 0.225 | 0.081 | 0.165 | 0.214 | 0.292 | 0.430 | 36.09 | 0.06 |
| 3 | Tetraodontiformes | 1 | 0.308 | 0.308 | 0.308 | 0.000 | 0.308 | 0.308 | 0.308 | 0.308 | 0.00 | 0.00 |
| 4 | Beloniformes | 1 | 0.037 | 0.037 | 0.037 | 0.000 | 0.037 | 0.037 | 0.037 | 0.037 | 0.00 | 0.00 |
| 5 | Pleuronectiformes | 3 | 0.141 | 0.350 | 0.269 | 0.091 | 0.228 | 0.315 | 0.333 | 0.350 | 33.99 | 0.05 |
| 6 | Clupeiformes | 16 | 0.118 | 0.426 | 0.282 | 0.075 | 0.244 | 0.265 | 0.327 | 0.426 | 26.60 | 0.04 |
| 7 | Aulopiformes | 3 | 0.180 | 0.254 | 0.221 | 0.030 | 0.204 | 0.228 | 0.241 | 0.254 | 13.80 | 0.02 |
| 8 | Scorpaeniformes | 1 | 0.273 | 0.273 | 0.273 | 0.000 | 0.273 | 0.273 | 0.273 | 0.273 | 0.00 | 0.00 |
| 9 | Carcharhiniformes | 5 | 0.021 | 0.214 | 0.109 | 0.066 | 0.072 | 0.097 | 0.143 | 0.214 | 59.96 | 0.04 |
| 10 | Rajiformes | 1 | 0.122 | 0.122 | 0.122 | 0.000 | 0.122 | 0.122 | 0.122 | 0.122 | 0.00 | 0.00 |
| 11 | Decapoda | 19 | 0.160 | 0.512 | 0.338 | 0.095 | 0.296 | 0.339 | 0.395 | 0.512 | 28.02 | 0.05 |
| 12 | Stomatopoda | 1 | 0.339 | 0.339 | 0.339 | 0.000 | 0.339 | 0.339 | 0.339 | 0.339 | 0.00 | 0.00 |
| 13 | Veneroida | 1 | 0.253 | 0.253 | 0.253 | 0.000 | 0.253 | 0.253 | 0.253 | 0.253 | 0.00 | 0.00 |
| 14 | Ostreoida | 1 | 0.150 | 0.150 | 0.150 | 0.000 | 0.150 | 0.150 | 0.150 | 0.150 | 0.00 | 0.00 |
| 15 | Arcoida | 1 | 0.138 | 0.138 | 0.138 | 0.000 | 0.138 | 0.138 | 0.138 | 0.138 | 0.00 | 0.00 |
| 16 | Sepiida | 4 | 0.214 | 0.245 | 0.230 | 0.013 | 0.220 | 0.230 | 0.240 | 0.245 | 5.53 | 0.01 |
| 17 | Teuthida | 1 | 0.205 | 0.205 | 0.205 | 0.000 | 0.205 | 0.205 | 0.205 | 0.205 | 0.00 | 0.00 |
| 18 | Neogastropoda | 3 | 0.079 | 0.232 | 0.159 | 0.063 | 0.123 | 0.167 | 0.199 | 0.232 | 39.40 | 0.04 |
| | Total/Mean | 133 | 0.021 | 0.512 | 0.241 | 0.095 | 0.178 | 0.227 | 0.315 | 0.512 | 39.46 | 0.07 |

QD–Quartile deviation, other abbreviations are similar to Table 1.

most sensitive attributes (Table 7). When we examined the difference in Euclidean distance between a full IRV and an abridged version of IRV (aIRV with 6 attributes), for 10 of the most resilient and vulnerable species, we found that the average change in Euclidean distance was 0.41 in the case of resilience and 0.30 in the case of vulnerability which is less than 13% of the total rank (Fig 7). This indicates that aIRV can also work similarly as IRV in the absence of a full set of information.

## Comparison of IRV with other vulnerability assessments

The R and V scores for 11 species comprising teleosts, elasmobranchs, crustaceans and molluscs were compared with the Productivity and Susceptibility scores of PSA for the same species (Table 8). On average, there was an 18.4% difference (range: 0 to 64.86%) in productivity scores with that of R score. In 4 out of 11 species, the PSA scored less than the R score. In susceptibility score, the average difference with V scores was higher at 33.7% (range: 1.4 to 160%). In 8 out of 11 species, the PSA scored less than V scores. The PSA appeared to underestimate susceptibility for highly vulnerable tropical species, particularly for elasmobranchs.

FishBase [26] has reported the resilience and vulnerability of all (102) fish species (teleosts and elasmobranchs) tested for IRV analysis. According to the FishBase, 85.3% and 10.8% of the species are moderately and highly resilient, respectively (Table 9). On the other hand, applying the resilience score and status used by FishBase, it is calculated that 61.8% and 33.3% of the species tested for IRV were moderately and highly resilient, respectively. Though there are more number of highly resilient species in IRV analysis, there is a similarity in the results as a high percentage of fish species (>95%) were in medium and highly resilient categories in both the assessments. However, the vulnerability analysis presented a different picture between the two analyses. While 89% of fishes were in low to moderate vulnerability in FishBase analysis, no species qualified in these categories in IRV analysis (Table 10). According to FishBase,

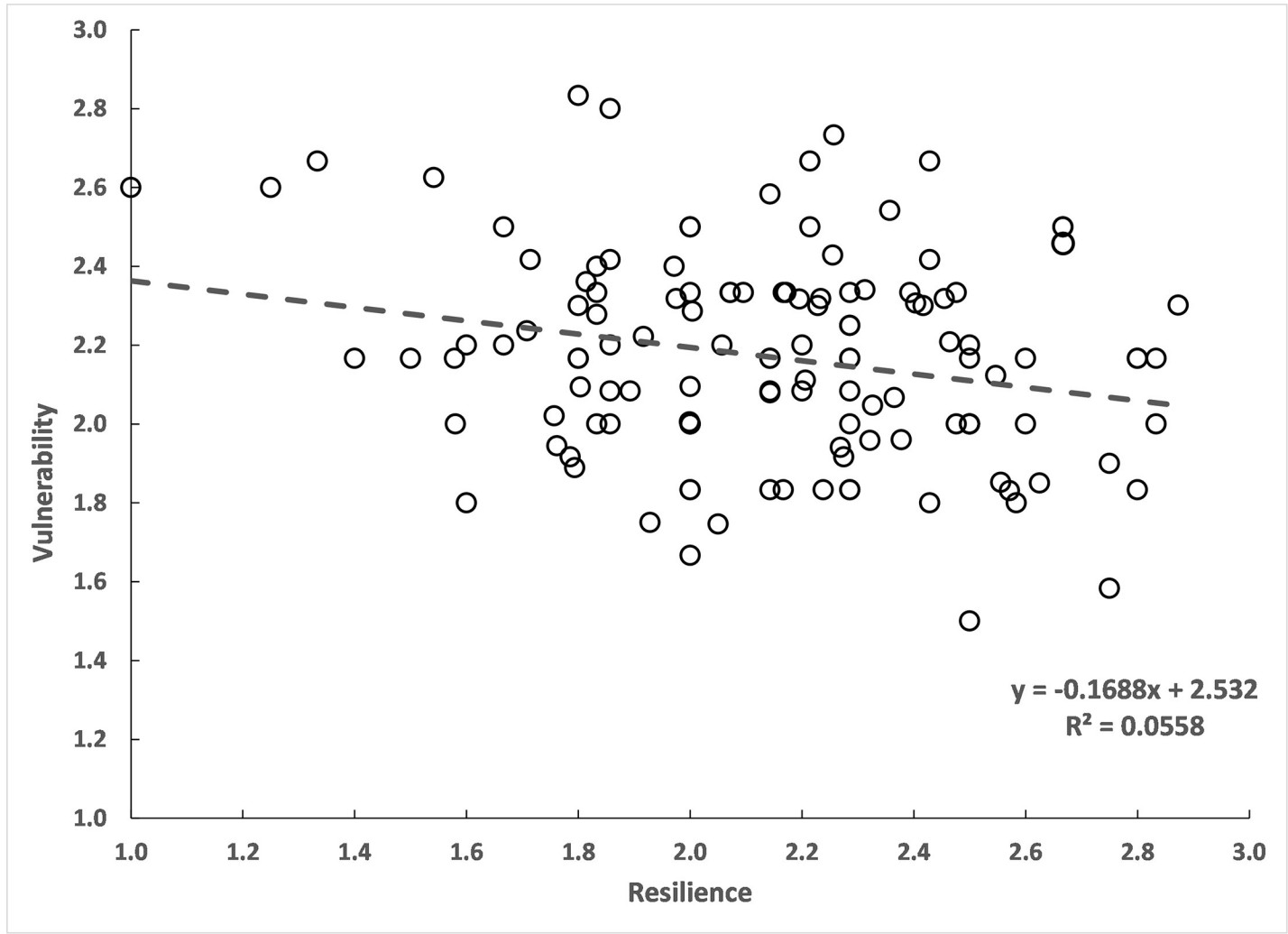

**Fig 5. Relationship between resilience score and vulnerability score; n = 133 species.**

only 11% of species were in high to very high categories, but if FishBase score-vulnerability status criteria are applied to IRV analysis, 100% of the species tested for IRV would be under high to very high categories. The criteria used by FishBase appears to underestimate the vulnerability of tropical stocks.

Of the 102 teleosts and elasmobranchs tested for IRV analysis, the IUCN has evaluated the status of 60 species for preparation of the Red List. Among the 60 species, 54 have been evaluated as 'least concern', two as 'vulnerable' and five as 'near-threatened. The two vulnerable species are G*laucostegus granulatus* and *Sphyrna lewini*, and the five near-threatened species are *Scomberomorus commerson, Thunnus albacares, Carcharhinus limbatus, Carcharhinus sorrah* and *Scoliodon laticaudus*. In IRV analysis, the first four species had low resilience ($< 2.0$), high vulnerability ($> 2.0$) and low IRV ($< 2.0$), showing an agreement with IUCN assessments.

## Discussion

The IRV analysis has given an opportunity to (i) assemble, synthesize and disseminate the best use of a large quantity of information on life-history traits and fishing impacts across species

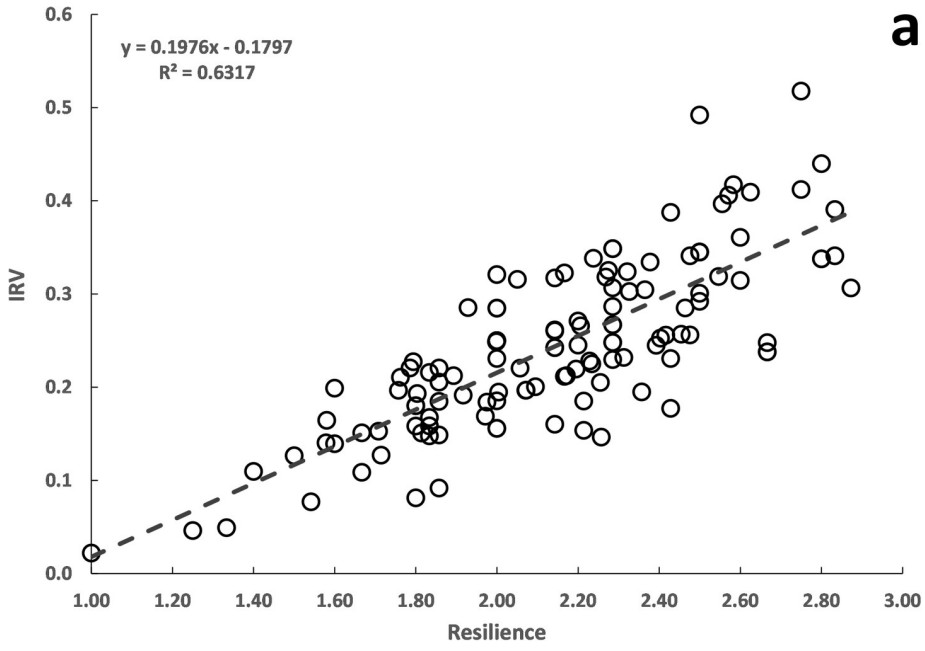

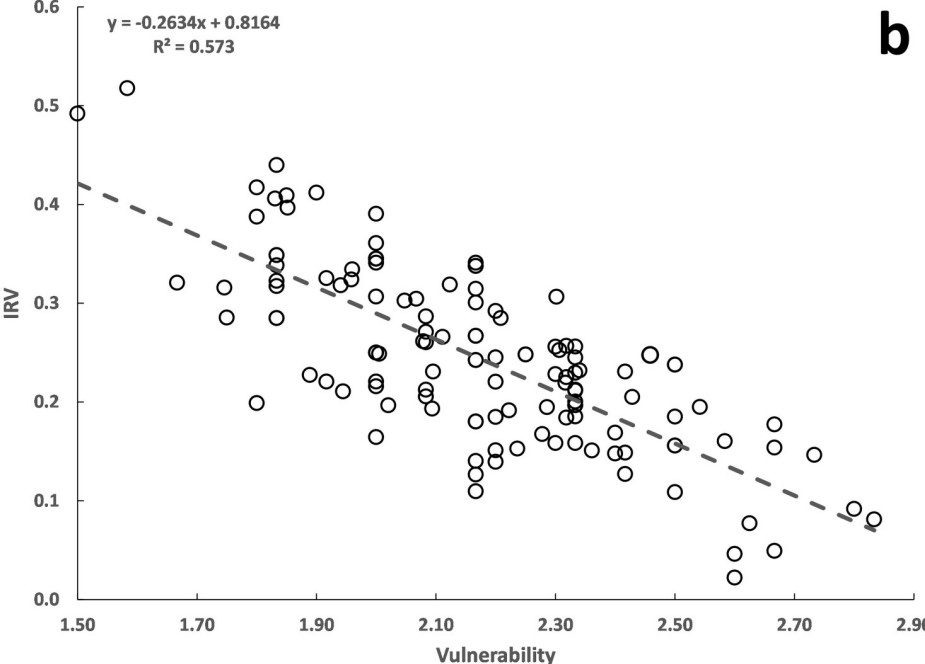

**Fig 6.** Relationship between resilience score and IRV (a) and relationship between vulnerability score and IRV (b); n = 133 species.

over a long period from a tropical region; (ii) identify gaps in data availability of a few important species or data-deficient species that are not represented in this analysis; and (iii) prioritise biological research needs to inform management and conservation of tropical fishes in the coming decades. As the source of data for the analysis is largely from commercial landings, it can be taken that the majority of the species covered in this paper contribute to fisheries.

**Table 7. Alexander's sensitivity analysis results scaled to unity showing the attributes ranked according to sensitivity.**

| Rank | Attribute | Alexander's S value |
|---|---|---|
| 1 | Exploitation ratio (ER) | 1.0000 |
| 2 | Growth coefficient (K) | 0.7285 |
| 3 | Asymptotic length (L∞) | 0.7274 |
| 4 | Coastal productivity index (CPI) | 0.5228 |
| 5 | Geographic distribution (Dist) | 0.5056 |
| 6 | Number of spawning months (NSM) | 0.5050 |
| 7 | Fecundity (FC) | 0.5017 |
| 8 | Body length depth ratio (BLD) | 0.4908 |
| 9 | Mean trophic level (MTL) | 0.4866 |
| 10 | Gear susceptibility (G) | 0.4751 |
| 11 | Length at recruitment (Lr) /L∞ ratio | 0.4444 |
| 12 | Length at maturity (Lm)/L∞ ratio | 0.4125 |

Rank 1 is the most sensitive attribute. Price attribute was not used in this analysis as they were direct input ranks.

While some species are targeted by a variety of craft-gear combinations, the remaining are non-targeted, but all species are important from a fisheries management perspective. The 133 species covered in this study belonged to 32 families of fishes; 9 families of crustaceans; 3 families of bivalves; and 2 families each of gastropods and cephalopods. Finfishes considered in this study contribute 6.1% and 15.6% to 1673 marine finfish species and 205 families, respectively recorded in the Exclusive Economic Zone of India (*www.fishbase.org*). It is observed that data is insufficient for a few important groups such as serranids (snappers, emperors), billfishes (swordfish, marlins), thresher sharks, rays, octopus and mussels that contribute a good share to the catches. These data-deficient groups are not represented in the IRV analysis as species-level data were available only on a few attributes. We suggest a large range of biological studies on these missing groups to fill the life-history data gaps. Besides, we have conceived the INMARLH database as a living and growing database with scope for additions and improvement, thereby enabling IRV analysis on more species.

## Data analysis

In this analysis, the criteria fixed for cutoff in each attribute have played the chief role in determining the R and V scores. Cutoff for each attribute was fixed based on the range of values for all species and segregating the range into three ranks from 1 to 3. For instance, the scoring for growth coefficient (K) was made by dividing the highest K observed in the study by 3, thereby dividing the scoring into equal thirds. Thus, the rank determined for each species was relative to that of other species and should not be viewed as an independent rank of each species. The scoring method is similar to that defined by [8] and [11].

The data source for the analysis were collected and published by different researchers during different years covering a long period. We noticed that the methodology of data collection and analysis of different attributes, fortunately, remained almost uniform across the researchers and periods. For example, annual growth coefficient, fecundity and exploitation ratio (Er) were estimated following length-frequency method, fecundity from *in situ* estimates on the number of eggs in mature and ripe females, and total mortality (Z) by length converted catch curve method and natural mortality by empirical relationship. While the methodology used was consistent among the researchers, it may be noted that the data on the attributes were

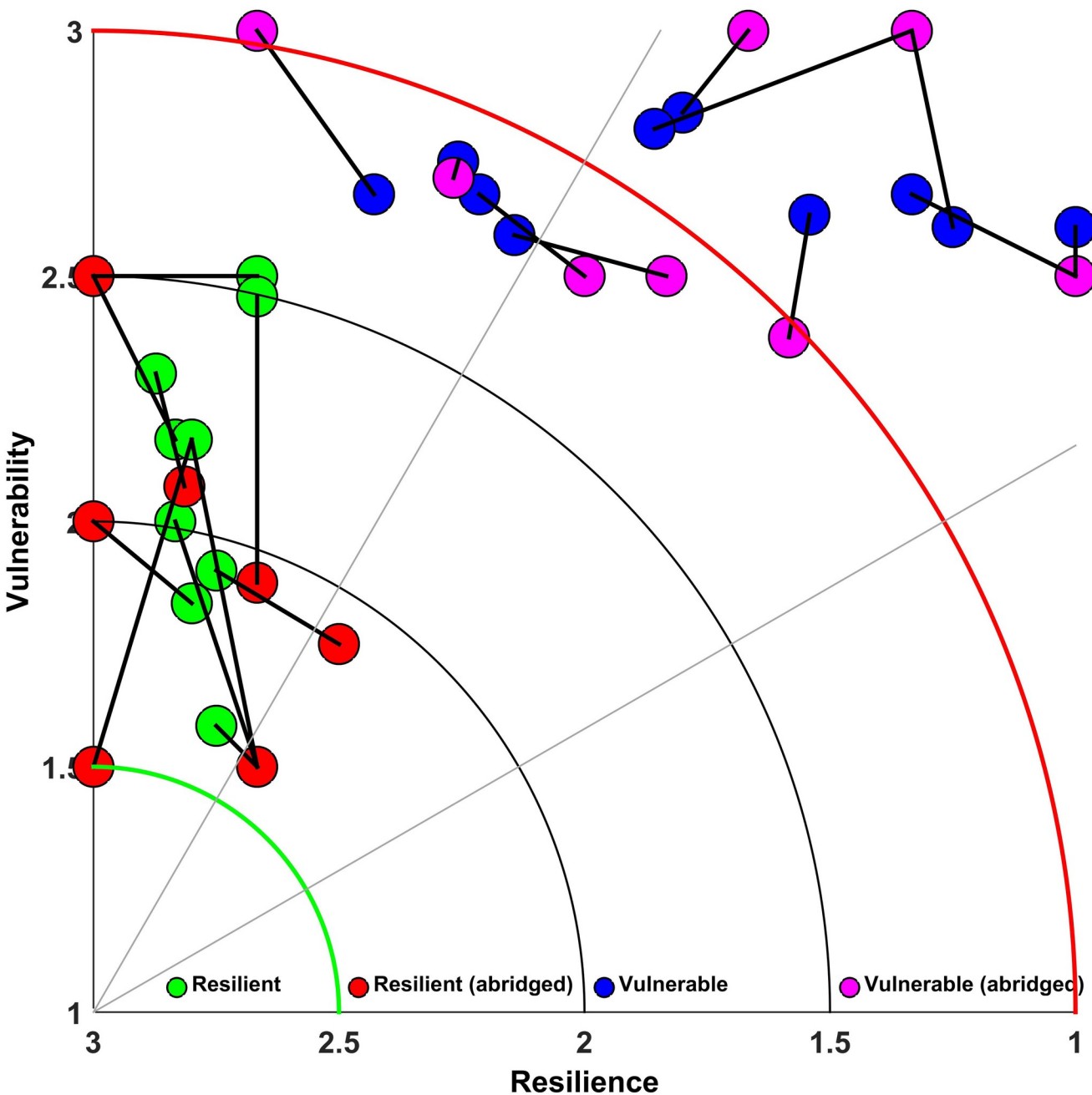

**Fig 7. Scatter plot showing the relative change in Euclidean distance of 10 most resilient and vulnerable species by applying the abridged (6) attributes.** Lines between points indicate the distance and angle of change from the original score.

collected during different periods. Due to this, temporal differences in IRV is a possibility, particularly in the attributes related to fishing over the years. For example, fishing effort and efficiency increased over the last three decades along the Indian coast, and consequently, it is expected that fishing mortality and thereby Er would be higher in the later years. However, a scrutiny of data used for the analysis showed that as most of the Er estimates were made in later years, the temporal trend in the Er values was not discernible. Hence, temporal differences were not expected to interfere with IRV analysis.

**Table 8. Comparison of PSA and IRV scores of 11 selected species.**

| Species | P of PSA | R Score | Difference | Percent change | S of PSA | V Score | Difference | Percent change |
|---|---|---|---|---|---|---|---|---|
| *Ablennes hians* | 1.00 | 1.25 | -0.25 | 25.00 | 1.00 | 2.60 | -1.60 | 160.00 |
| *Alepes djeddaba* | 2.73 | 2.43 | 0.30 | 10.99 | 1.86 | 2.42 | -0.56 | 30.11 |
| *Alepes klenii* | 2.56 | 2.14 | 0.42 | 16.41 | 2.14 | 2.17 | -0.03 | 1.40 |
| *Arius jella* | 1.67 | 1.86 | -0.19 | 11.38 | 2.29 | 2.00 | 0.29 | 12.66 |
| *Carcharhinus limbatus* | 1.11 | 1.83 | -0.72 | 64.86 | 2.14 | 2.33 | -0.19 | 8.88 |
| *Parapenaeopsis stylifera* | 2.56 | 2.56 | 0.00 | 0.00 | 2.29 | 1.85 | 0.44 | 19.21 |
| *Uroteuthis duvaucelli* | 2.56 | 2.03 | 0.53 | 20.70 | 1.71 | 2.20 | -0.49 | 28.65 |
| *Portunus pelagicus* | 2.67 | 2.48 | 0.19 | 7.12 | 2.14 | 2.33 | -0.19 | 8.88 |
| *Sepia pharaonis* | 2.33 | 2.22 | 0.11 | 4.72 | 2.25 | 2.20 | 0.05 | 2.22 |
| *Sphyrna lewini* | 1.36 | 1.67 | -0.31 | 22.79 | 1.57 | 2.50 | -0.93 | 59.24 |
| *Carcharhinus sorrah* | 1.22 | 1.00 | 0.22 | 18.03 | 1.86 | 2.60 | -0.74 | 39.78 |
| **Average** | | | | **18.4** | | | | **33.7** |

P indicates the productivity score of PSA and S indicates the susceptibility score of PSA. R and V scores are based on the current method. The percentage difference is based on absolute values.

The IRV analysis is based on the premise that resilience and vulnerability are heterogeneous between fish species as there are differences in life-history traits and the impact of fishing. The IRV has been estimated as a combination of the two groups of attributes and consequently, IRV is expected to differ between the species. However, a species with a low level of resilience would not necessarily be at high risk (low IRV) unless it is impacted by fishing. This result is similar to the PSA reported by [11], in which stocks with a low level of productivity was not vulnerable to fishing unless there was also susceptibility of the stock to the fishery.

Data on each attribute cover a wide range; for example, the annual growth coefficient ranged from 0.12 (*Plicofollis dussumieri*) to 3.9 (*Acetes indicus*); fecundity from 1 (*Glaucostegus granulatus*) to 4.7 million (*Panulirus polyphagus*); exploitation rate from 0.12 (*Sardinella gibbosa*) to 0.93 (*Netuma thalassina*). Large body size, late age at maturity and low fecundity are considered life-history traits that increase the vulnerability of species to fishing. Therefore, it is no surprise that the list of ten most vulnerable species includes sharks, seerfish and barracuda. On the other hand, the ten most resilient species consists exclusively of small-sized, highly fecund and low trophic level species of 5 crustaceans, 4 teleosts and one gastropod. The disparities in IRV among species within the same family indicates the difference in the biological characteristics and response to fishing by the species within the same family. The family Scombridae consists of Indian mackerel and coastal and oceanic tunas that greatly differ in body size, length-at-maturity, distribution, exploitation rate and price.

The capacity to recover from impacts depends on the severity of the impact and the inherent resilience capacity. While demographic attributes such as longevity, growth rate, fecundity,

**Table 9. Comparison of FishBase [26] and IRV analysis on resilience score for 102 teleost fish species.**

| Resilience score | Status | FishBase analysis | | IRV analysis | |
|---|---|---|---|---|---|
| | | # of species | % | # of species | % |
| 0–0.75 | Very Low | 0 | 0 | 0 | 0.0 |
| 0.76–1.5 | Low | 4 | 3.9 | 5 | 4.9 |
| 1.51–2.25 | Medium | 87 | 85.3 | 63 | 61.8 |
| 2.26–3 | High | 11 | 10.8 | 34 | 33.3 |

**Table 10. Comparison of FishBase [26] and IRV analysis on vulnerability score for 102 teleost fish species.**

| Vulnerability | FishBase analysis | | | IRV analysis | | |
|---|---|---|---|---|---|---|
| | FishBase score | # of species | % | IRV score | # of species | % |
| Low | 0–25 | 43 | 42.2 | 0–0.75 | 0 | 0 |
| Moderate | 26–50 | 48 | 47 | 0.76–1.5 | 0 | 0 |
| High | 51–75 | 5 | 4.9 | 1.51–2.25 | 65 | 63.7 |
| Very high | 76–100 | 6 | 5.9 | 2.26–3 | 37 | 36.3 |

recruitment and natural mortality determine the productivity of a species [9], IRV analysis shows that among the 7 tested resilience attributes, favourable L∞, Lm and fecundity contributed to higher resilience of stocks. This is evident from a large number of species that had resilience score above 2.5 in these 3 attributes; of the total number of species tested for L∞, Lm and fecundity, 64.7%, 58.0% and 58.0% respectively had scored above 2.5. Small-sized fishes and crustaceans are highly productive with high fecundity and faster turnover of generations. For example, L∞ of oil sardine *Sardinella longiceps* is 23.4 cm, Lm is 63.8% of Lmax, *in situ* fecundity ranges from 60,807 to >100,000 and the turnover of generation is 1 year [34]; the L∞ of *Metapenaeus affinis* is 21.5 cm, Lm is 43.7% of Lmax, *in situ* fecundity ranges from 73,500 to 115,600 and the turnover of generation is < 1 year [35]. Despite severe fluctuations in productivity over the years, the landings of small pelagics and crustaceans together contributed as high as 29.1% to the total landings along the Indian coast during 2016–2018 [35], reflecting the high resilience of the species.

Many species are vulnerable due to unfavourable body length-depth ratio (BLD: 79.8% of the species tested had ratio < 5) and length-at-recruitment (46.0% of the species had Lr/L∞ ratio < 0.3). While the gear type (such as trawl or gillnet) did not drive many species to vulnerability (gear susceptibility), the gear specifications, particularly the mesh size appears to have impacted the species. The presence of a large number of species with low BLD (< 5) and Lr/L∞ ratio (< 0.3) in the high vulnerability group shows that the mesh size of nets is small, and thereby, the probability of escapement through nets is low and fishes are recruited into the fishery at a small size. The capture of juveniles has emerged as a major issue in the marine fisheries in India. For example, exploitation of juveniles is substantial, contributing 21% to the trawl catch along the Mangalore coast (an important fishing centre along the southwest coast [36]. The codend mesh size of trawls is very small (15 to 20 mm) despite the recommended size of 35 mm, which is a serious concern for the sustenance of the fishery.

The weak relationship between R and V maybe because of the following reasons: (i) For many species that are clustered around a given R, the V varied widely. For example, for R = 2.0, the V of nine species ranged from 1.67 (*Secutor insidiator*) to 2.50 (*Cynoglossus arel* and *Placuna placenta*). The pattern is similar for species with very high R (> 2.5). (ii) For many species that had high R, the V was also high. For example, for R = 2.67, the V was 2.5 (*Lactarius lactarius*, *Gazza minuta* and *Babylonia zeylanica*). (iii) Only a few species showed clear expected scores of high R and low V, or low R and high V. These trends indicate that all species with favourable biological traits need not be in the safe zone but could be impacted by fishing. For example, high fecundity does not ensure a low exploitation ratio or reduction in gear susceptibility. (iv) Two V attributes, namely gear susceptibility and price of species are not connected to any of the chosen R attributes. Hence, these two attributes behave independently of the R attributes.

The IRV is jointly determined by the R and V scores, and higher the IRV, the species is expected to be exposed to low risks. As a group, the elasmobranchs (IRV: 0.120) and crustaceans (0.340) are at two extremes of the spectrum, with the elasmobranchs at the risky end of

the IRV. Unlike the non-existence of a relationship between the R and V, clear relationships were evident between R and IRV as well as V and IRV. With increasing R, the IRV increased and with increasing V, the IRV decreased. Despite the existence of linear relationships, deviations from the trendlines were discernible. For example, in the R vs IRV relationship, deviations were observed in the IRV particularly in higher R values, as in the case of *Nematopalaemon tenuipes* and *Solenocera crassicornis*. These two species had high resilience scores (2.50–2.75), and as they are not the main target, the vulnerability scores of these species are the lowest (1.50–1.58) among the tested species. In the V vs IRV relationship, for the given V = 2.17, the IRV ranged from 0.110 (*Eupleurogrammus muticus)* to 0.341 (*Sardinella fimbriata*). Large differences in the life history characteristics among the species with the same IRV have influenced the deviations from the linear relationship between V and IRV.

The top-10 risky species from the IRV list clearly show that except the chank *Turbinella pyrum*, all others are large-sized fish species with high trophic levels. Among the top-10 risky species, the top-6 have a high V score. The remaining 4 species (*Sphyrna lewini*, *Eupleurogrammus muticus*, *Otolithoides biauritus and Glaucostegus granulatus*) in the IRV list have qualified as high-risk species as they are less vulnerable compared to the first 6 species, but are relatively low in resilience. Although theoretically, the IRV of a highly resilient species can have a score of 1, the maximum observed was only a little more than half of this (0.512). This may indicate that tropical marine species are resilient-yet-vulnerable, a mixed feature characteristic of complex ecological systems where networks were far more robust to historical conditions than new ones [37]. This can indicate the heightened vulnerability of tropical marine species to newly emerging climate changes.

## Sensitivity analysis

Sensitivity analysis determines how the variables are affected based on changes in input values and it has been used effectively for PSA analysis by many researchers (for example, [38]). Of the 13 tested attributes in IRV analysis, four resilience and two vulnerability attributes were found to be sensitive and influential in determining the R and V scores and thereby the IRV (Table 6). As the average change in Euclidean distance is negligible if only these six attributes are used for analysis, an abridged version of IRV (aIRV) is as effective as the longer version. The sensitivity analysis shows that not all attributes are equally important in determining the IRV of a species. Some versions of PSA used an attribute weightage system in which higher weights were applied to the more influential attributes [8,10,39]. However, we did not assign weightage to attributes as it would make regional comparisons difficult. The importance of the six influential attributes are narrated below:

Exploitation ratio (Er) has been assessed as the most sensitive attribute. Er is a quantitative measure that gives a snapshot of species status. It helps to identify species that experience high fishing mortality and to take appropriate management measures. In IRV analysis, 42% of species had high Er (vulnerability score: > 2.0; Fig 1B). In multigear fisheries, the Er for the same stock may vary between two gear types because they will cause different fishing mortalities. Ideally, each fishery should have its vulnerability evaluation performed to determine which stocks in that fishery are most vulnerable [11]. Calculation of Er by gear type would provide more useful information for developing management measures.

The next influential attributes are two highly correlated resilience attributes, namely growth coefficient and asymptotic length. The von Bertalanffy growth coefficient (K) measures how rapidly a fish reaches its maximum size. Many small-sized groups such as clupeids and crustaceans fall under the high K category [40]. These species also represent key energy pathways from planktonic communities to higher trophic level predators such as sharks, tunas,

barracudas, cetaceans, etc. Commercial fisheries for these stocks are typically high-volume and thereby constitute economically important fisheries. The fishery for clupeids such as the sardines and anchovies are highly dynamic in space and time with tremendous interannual variability in productivity and abundance. The clupeids are targeted by small-meshed gillnets and ringseines, and crustaceans by trawls.

The fourth influential attribute, namely, the Coastal productivity index (CPI) is linked to the fifth attribute, the species geographic distribution (Dist). While species maximally distributed in highly productive waters are considered more resilient, species which have narrow geographic distribution are more vulnerable than species with a relatively wider distribution. Species that are distributed in proportionately larger areas of high CPI (such as upwelling areas or with high chlorophyll concentration) have higher productivity and abundance as demonstrated by trophic flow models [41]. Species with larger distributional areas can adapt to differential environmental conditions and are therefore more resilient [9,11,15]. Contrary to this, macroecological theory predicts that geographic range is positively related to species with maximum body size and higher mobility explaining their larger range size [42], and suggesting that fish species with a large geographic range may be more vulnerable to fishing. However, the majority of species considered for IRV analysis are not those with higher mobility but with localised distribution spread over larger areas; for example, the Indian mackerel *Rastrelliger kanagurta* and threadfin bream *Nemipterus japonicus* which do not undertake large-scale migration, but are distributed along the entire Indian coast. Since they form large populations along the entire coastline, they are more resilient than the species with restricted distribution. In a similar vein, higher ecosystem resilience to disturbances was observed in coral reefs which were geographically interconnected [43].

The sixth influential attribute, the number of spawning months, considers that prolonged annual spawning for 6 or 7 months or more enables the species to release of eggs and larvae for a longer duration, thereby enhances their adaptation and survival to temporal environmental variations. Contrary to this, the chance of survival of eggs and larvae of species with shortened spawning duration of one or 2 months is jeopardized in the event of higher predation or unfavourable environmental condition in a given year, thereby negatively affecting their abundance.

Recalculating of PSA vulnerability scores with 3 to 5 attributes and comparing with prediction error rate with that of 12 attributes in the PSA developed by [11], revealed that the highest prediction accuracy occurring when only one productivity and two susceptibility attributes were used and the lowest accuracy when all 12 attributes were used [7]. Though six attributes have been identified as playing important roles in determining the resilience and vulnerability in IRV analysis, reproducing and restricting the application of only these six sensitive attributes to other fisheries have to be exercised with caution. In many earlier PSA analyses, the length at maturity (Lm)/L∞ ratio has been identified as the most significant predictor of risk. However, this attribute has been relegated to a lower level of importance in IRV analysis. Considering that the qualitative frameworks are often subjective, consolidating experts' knowledge is important for reproducing the results of the analysis to other fisheries.

## IRV and other vulnerability assessments

Comparison of IRV with PSA [8], FishBase [26] and IUCN [44] evaluations shows contrasting results in the assessments of resilience as well as vulnerability, but more conspicuously on vulnerability. The following reasons are discernible for the differences: (i) The objectives of the assessments differed between the analyses. While the objective of IRV analysis is to assemble the available data and develop a broad methodology to examine the likely impact of fishing on

multispecies tropical species, the PSA is aimed to examine the likely impact of trawling on the sustainability of teleost bycatch species [8]. The Ecological Risk Assessment for Effect of Fishing (ERAEF), which is included in PSA, is a hierarchical framework for a comprehensive assessment of the ecological risks arising from fishing, with impacts assessed against five ecological components–target species; byproduct and bycatch species; endangered, threatened and protected (ETP) species, habitats, and (ecological) communities [10]. Vulnerability and Resilience analysis by FishBase [26] is aimed at understanding the effect of fishing with applications such as Intrinsic Vulnerability analysis to test if fish species adapted to different habitats have different vulnerabilities to fishing; and if changes in the species composition of catches were related to the intrinsic vulnerability of the exploited stocks [15]. The Red List of IUCN also assesses the vulnerability of fish species, with the main objective of identifying risks to biodiversity. It assesses the endangerment of a species but allows assessment of the vulnerability of only a limited number of well-studied species [45]. The IUCN assessment is extremely useful to design a protection plan for a specific species which is being endangered. However, as the purpose of species evaluation by the IUCN is the preparation of a Red List for conservation purpose, 90% of the species evaluated have been categorised as 'least concern. Considering the objective of IRV and the objective and classification criteria used by IUCN, the results from the two analyses are very different. Corresponding to differences in the objectives, the approach to analyse resilience and vulnerability of fish stocks varied, and as a result, all these methods do not convey the same information. (ii) The type of attributes selected had large variations. Among the 13 attributes used by [8], only one productivity attribute (termed by the authors as 'recovery' attribute) and one vulnerability attribute (termed as 'susceptibility' attribute) matched with the attributes used for calculating IRV—maximum body size (among recovery attributes) and species distributional range (among susceptibility attributes). Moreover, the analysis by [8] applied a weightage score for 10 of 13 attributes, whereas all attributes were treated with equal importance for IRV analysis. Modified versions of PSA [10,11] used 22 and 21 attributes, and among them, only 9 attributes are similar to the attributes used in IRV analysis. Thus, the reliance on the number and types of attributes differed between the analyses. (iii) The scoring thresholds (cutoff) for the attributes used by PSA and FishBase are based on qualitative extinction risk assessment [24] and the PSA of [8]. The scoring thresholds have been modified in some analyses to better suit the distribution of life-history characteristics of specific cases (for example, PSA developed for USA fisheries by [11]). Using scoring threshold different from temperate species is inevitable for tropical species. In the PSA, for example, the maximum body size of fish measuring < 813 mm, 813–1066 mm and > 1066 mm were categorised as high, medium and low resilience, respectively. In the IRV, as many species are smaller in size in the tropical fisheries, < 400 mm, 401–800 mm and > 800 mm were categorised as high, medium and low resilience, respectively. Thus, a species with a maximum size of 810 mm was highly resilient in the PSA whereas it was of low resilience in the IRV. As the cutoffs used for categorizing stocks into high, moderate and low ranks under different productivity and susceptibility attributes do not align with many other life-history characteristics (such as fecundity, asymptotic length, etc) of tropical stocks, a different set of cutoffs was applied for IRV analysis. Due to these differences in approach, great care should be taken for reproducing these frameworks to other fisheries (see also [7]).

## Utilization of IRV results

The development of IRV has allowed the sorting of species into low, medium and high vulnerability for considering the impact of fishing, enabling utilization of the results for better management of fisheries resources and as an input in novel stock assessment methods [46]. While

the species/fisheries with high vulnerability warrant a detailed look at an action plan for management or conservation, the analysis is also a tool for rapidly assessing the vulnerability of stocks leading to assessments or management actions. The species screened as highly vulnerable and risky deserve to be prioritized for quantitative analysis. From the analysis, 14 species are in the list of top-10 vulnerable as well as high-risk species and 4 are in the list of high-risk species alone. A positive aspect in marine fisheries research in India is that information on attributes other than those selected for IRV analysis such as spawning stock biomass, Maximum Sustainable Yield and abundance estimates are available for many species. Though these quantitative measures are not available on a year-to-year basis, they are good enough to be used as complementary data, particularly on vulnerable and high-risk species. For example, the MSY, spawning stock biomass (SSB) have been estimated for the milk shark *Rhizopriono-don acutus* and it has been shown that the prevailing exploitation rate has reduced the virgin stock biomass ($B_o$) and SSB to 55% and 34%, respectively [47]. While IRV alone cannot identify management options, it can provide insights into developing effective management measures in combination with other complementary information.

The Wildlife (Protection) Act 1972 of India has listed 10 species of elasmobranchs, 24 teleosts, 20 bivalves and 4 gastropods in addition to > 800 species of corals, sponges, sea cucumbers, turtles and marine mammals for protection under different Scheduled Lists. While the listed species are endangered and the Act provides protection from hunting and trade, there is no provision to list vulnerable species contributing to fisheries. The species identified as high risk and highly vulnerable may be considered for listing and developing management plans for increasing the probability that overfishing does not occur. It is also important to consider rebuilding protocols needed for these species.

Marine fisheries management in India is in a state of regulated open access. Seasonal fishing closure is an important control measure followed to protect the spawning stocks. The Marine Fishing Regulation Act (MFRA) has provision for other input control measures such as regulating the number of boats and mesh size, but the implementation of these measures need improvement. Output control measures are absent, except implementation of minimum legal size at capture in a few fisheries. In the existing fishing regulation, a specific management plan does not exist for high risk and high vulnerability species. Fishery Management Plan (FMP) exists for only a few specific fisheries such as short-neck clams, elasmobranchs and blue swimming crabs [48–50]. The species identified under high risk and high vulnerability categories in the IRV analysis may be considered as candidates for developing Fishery Management Plans.

The Marine Stewardship Council suggests that susceptibility scores in the PSA can provide the basis for management actions to reduce risk from fishing [22]. For a species showing high risk or high vulnerability in IRV analysis, for example through a short spawning period, the seasonal closure of fishing, that is already in place in India, may consider revising the peak spawning months of that species for closure. Similarly, actions on closed areas (to reduce fishing in areas of abundance of the species), or gear modifications (stringent implementation of mesh size regulations to improve length-at-capture and reduce low-value bycatch) or regulating fishing effort (to reduce fishing mortality) which would reduce the vulnerability scores and improve the IRV score. These actions may have consequences for other species or fisheries as reducing vulnerability scores will reduce risk. On the other hand, resilience is inherent to the species, and change in fishing practices cannot change the scores of the R attributes. However, in species like the catfish *Nemapteryx nenga*, a moderate IRV (0.199) masks the low R (1.60) because of low fecundity and higher trophic level (3.9) in addition to narrow distribution that increases vulnerability. Ignoring this type of species from protection is likely to increase their vulnerability and risk.

## Conclusions

In the current analysis, the IRV of different species have been estimated and projected on a Pan-India level. This macro-level approach is likely to mask the vulnerability and risk factors of specific fisheries within narrow and unique geographical ranges. As the data for IRV calculation were collected from a vast geographical scale ranging from 8°N to 23°N along the east (Bay of Bengal) and west coasts (Arabian Sea) of India, there is a possibility of spatial differences in the IRV between different fisheries. For example, the species composition of the bottom trawl fishery, which is ubiquitous along the entire Indian coast, varies greatly between different zones. On the other hand, while pelagic fishery for the oil sardine *Sardinella longiceps* occurs in many areas, it occurs predominantly along the southwest and southeast coasts using the ring seines. Another example is the dolnet fishery for the Bombayduck *Harpadon nehereus* that is unique along the northwest coast. As the next step, IRV calculation for India may focus on the assessment of specific fisheries. The recent suggestion on zonation of the Indian Exclusive Economic Zone into 13 territorial water zones (area within 12 nautical miles from the shore) and 6 regional zones (areas between territorial waters and EEZ boundary) for stock assessment and fisheries management [51] appears to be an ideal platform for calculating IRV for specific fisheries in these zones.

In the multispecies and multigear fisheries that are prevalent along the Indian coast and in other tropical seas of the world, the IRV calculation for specific fisheries may be done to calculate cumulative IRV of all major fisheries in a location by following an appropriate tool [5] for simultaneously assessing multiple, geographically co-occurring fisheries and establishing priorities for monitoring, management, and conservation efforts. This study shows the importance of accounting for the potential cumulative impacts of multiple co-occurring fisheries for the conservation of coastal marine ecosystems, identifies relative risk imposed by multiple fisheries, and provides a tool for a preliminary evaluation of the possible outcomes of management alternatives. This approach would enable operationalizing the concepts of the Ecosystem Approach to Fisheries Management (EAFM) as well. One of the key goals of EAFM is to assess and manage the cumulative impact of multiple fisheries, both on the species targeted by the fisheries and on the ecosystem, including non-target species.

For calculation of IRV for specific fisheries, the R scores in the present analysis, as they represent life-history traits, could be used for the same or related species. However, fresh data may have to be collected for species that are not covered in the present analysis. The V scores will differ for each fishery as they are subject to the types and intensity of fishing practices. As species from the same or similar families in a fishery may have quite different vulnerabilities [39], grouping species by vulnerability rather than taxonomically could lead to a more precise group with the most appropriate indicator species [52]. In other words, we recommend that fisheries management experts may determine management measures for highly vulnerable and high-risk stocks on a fishery-by-fishery basis. In a broader sense, the IRV analysis has given an insight into ways for better management by expanding and validating the analysis for specific fisheries.

## Supporting information

**S1 Table. List of sources used for collating life history and fishery parameters of tropical marine species of the Indian subcontinent.**
(DOCX)

**S2 Table. List of 133 species (current nomenclature as per WoRMS database available from http://www.marinespecies.org) used in the IRV assessment.** Due to recent taxonomic revisions in some cases, the old nomenclature is also shown in brackets along with common

names and the family of the species. Primary classification into 6 broad groups (teleosts, elasmobranchs, crustaceans, bivalves, cephalopods and gastropods). Further classification of the list is family-wise, alphabetically. The species numbers are used in figures (resilience and vulnerability plots). The broad habitat of the species is indicated by pelagic (P), demersal (D) and midwater (M).

(DOCX)

## Acknowledgments

The authors are grateful to the Director of ICAR-CMFRI, Kochi for facilities. We thank the FAO for the encouragement to take forward the initial results of this work presented in an FAO workshop in Bangkok, Thailand 10 years ago. We are grateful to Dr Shelton Padua for rendering the map shown in Fig 1. We thank 3 anonymous reviewers for critical comments which greatly improved the manuscript. We are also grateful for technical assistance from the staff of the FRA division of CMFRI.

## Author Contributions

**Conceptualization:** Kolliyil S. Mohamed, Elayaperumal Vivekanandan.

**Data curation:** Thayyil Valappil Sathianandan, Somy Kuriakose, U. Ganga, Saraswathy Lakshmi Pillai, Rekha J. Nair.

**Formal analysis:** Kolliyil S. Mohamed, Thayyil Valappil Sathianandan, Elayaperumal Vivekanandan, Somy Kuriakose, U. Ganga, Saraswathy Lakshmi Pillai, Rekha J. Nair.

**Investigation:** Kolliyil S. Mohamed, Somy Kuriakose, U. Ganga, Saraswathy Lakshmi Pillai, Rekha J. Nair.

**Methodology:** Thayyil Valappil Sathianandan, Elayaperumal Vivekanandan, Somy Kuriakose.

**Project administration:** Kolliyil S. Mohamed.

**Resources:** Kolliyil S. Mohamed, Thayyil Valappil Sathianandan, Elayaperumal Vivekanandan, U. Ganga, Saraswathy Lakshmi Pillai, Rekha J. Nair.

**Software:** Thayyil Valappil Sathianandan.

**Supervision:** Kolliyil S. Mohamed, Elayaperumal Vivekanandan.

**Validation:** Somy Kuriakose, U. Ganga, Saraswathy Lakshmi Pillai, Rekha J. Nair.

**Visualization:** Kolliyil S. Mohamed, Thayyil Valappil Sathianandan.

**Writing – original draft:** Kolliyil S. Mohamed, Elayaperumal Vivekanandan.

**Writing – review & editing:** Kolliyil S. Mohamed, Thayyil Valappil Sathianandan, Elayaperumal Vivekanandan.

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
