## [Decision Letter · Decision Letter 0]

9 Feb 2021

PONE-D-20-34956

Application of biological and fisheries attributes to assess the vulnerability and resilience of tropical marine fish species

PLOS ONE

Dear Dr. Sathianandan,

Thank you for submitting your manuscript to PLOS ONE. After careful consideration, we feel that it has merit but does not fully meet PLOS ONE’s publication criteria as it currently stands. Therefore, we invite you to submit a revised version of the manuscript that addresses the points raised during the review process.

Please address all reviewers' excellent comments and suggestions for improving your manuscript. And, please, proof read your ms to improve the language and presentation of your results. As indicated across all reviewers, the subject matter is of global importance and needs to be contextualized as such. Please support your writing with appropriate citations both in the introduction and discussion.

We look forward to receiving your revised manuscript.

Kind regards,

Ismael Aaron Kimirei, Ph.D.

Academic Editor

PLOS ONE

Journal Requirements:

Reviewers' comments:

Reviewer's Responses to Questions

**Comments to the Author**

1. Is the manuscript technically sound, and do the data support the conclusions?

Reviewer #1: Partly

Reviewer #2: Partly

Reviewer #3: Partly

2. Has the statistical analysis been performed appropriately and rigorously? 

Reviewer #1: No

Reviewer #2: No

Reviewer #3: I Don't Know

3. Have the authors made all data underlying the findings in their manuscript fully available?

Reviewer #1: No

Reviewer #2: No

Reviewer #3: Yes

4. Is the manuscript presented in an intelligible fashion and written in standard English?

Reviewer #1: Yes

Reviewer #2: No

Reviewer #3: No

5. Review Comments to the Author

Reviewer #1: 1) The manuscript is technically sound and the data support the conclusions. More could be done with the data, however, and the data set could be clearer. I could not access the attached data file, so I could not fully evaluate the data.

2) Very few statistics were presented. Most of the results were presented as categories of resilience or vulnerability. I suggest the authors consult biostatisticians to consider appropriate statistical tests.

3) No, I could not access the attached file.

4)The manuscript is written in standard English and with the exception of a relatively small number of sentences, was easy to read. I found the manuscript, however, to be overly verbose and repetitive. The manuscript was much longer than it needed to be and had too many small sections. It should be streamlined. Also, the authors spent too much time describing the data in the tables and figures. They would have a stronger manuscript if they spent more time interpreting their results instead of repeating what readers can see in a table or figure.

5) Additional statistics would help. For example, the authors describe the range of values they calculated, but it would be useful to know the values of each quartile of the data, or median, mean, and 95% confidence interval of the data. Statistical analyses could then be conducted to compare different taxa or fisheries. In the Discussion, the authors suggest that these types of analyses could be conducted on a regional basis, but it seems as though they have the data to do that. Why didn't they? A major comment I have is that these types of analyses have been conducted around the world, but there are very few references to indicate that the authors have read or considered other researcher's work.

The authors have clearly spent a great deal of time collecting data to evaluate the vulnerability and resilience of fishes in India. Their topic is of great interest to people all over the world. If they would delve more into the world literature and conduct some additional analyses, they could make a strong contribution to fisheries management. The Discussion section would be strengthened by a more detailed discussion of how their results could be applied to fishery management.

Reviewer #2: Overall comments:

Thank you for the opportunity to provide feedback. This manuscript was an interesting read and tackles an important issue for improving fisheries management of multiple species in Indian coastal waters. The authors demonstrate the need for such information in the introduction, and they explore the utility of using biological and fishery attributes to quantify the resilience and vulnerability to different fish species using three metrics. While I think the manuscript would be thematically appropriate for this journal, I think there are substantial issues with the level of detail and writing clarity provided in this draft, which makes it difficult to assess the authors’ approach. Thus, I cannot recommend this manuscript for acceptance in PLOS ONE at this time. I provide general and specific comments below to assist in efforts to improve the manuscript.

General comments:

- In general, this manuscript would benefit from a careful proofreading, as there are numerous spelling and grammatical mistakes throughout. I highlight some of these in the Specific comments below. This may help the writers clarify their meaning in some sections of the paper, which will be helpful for the overall interpretability.

- There are many statements throughout the introduction and discussion that would be much stronger with a supporting citation. I think this manuscript could do a more thorough job of incorporating relevant published work.

- I do not agree that resilience and vulnerability are innately opposites, as is hypothesized here. As indicated in L652-653, you expected high R with low V (or vice versa) but did not always observe this result. This does not surprise me, since the two indices you calculate (R and V), while complimentary, describe attributes of each species/fishery that are not necessary inversely related (or related at all). At points throughout the manuscript, it is implied that R and V are inversely related, and these statements need to be adjusted to either justify this expectation by the authors or to clarify their expectations if my interpretation above is incorrect.

- A figure showing a map of the study regions (data source areas) would be helpful and could be referenced throughout the manuscript to provide better spatial context (e.g., L192-193, L870-873)

Specific comments:

L22: delete “was estimated”.

L28: “large-sized” should be changed to “large”.

L30-31: delete “by the species within the same family”.

L33: rather that “contradictory results”, use something like “different species vulnerabilities” or “different fish management priorities”.

L47: “species of diverse” should be “species with diverse”.

L50: delete “number of”.

L89: both mentions of “is” should be changed to “are”.

L106-107: with regard to the Hilborn et al. (2020) citation, is there any information available to allow you to describe the state of the fisheries used in this study? Are they “in poor shape” as you say?

L115: BMSY and FMSY have not been defined elsewhere in the manuscript.

L119-127: This paragraph brings up an important point. You describe 133 species, but use data from 644 stocks. Are there multiple stocks for some of these species? If so, how were data from different stocks combined for a single species?

L141-143: although the original paper is cited, I believe it would be helpful to briefly describe the attributes from the siFISH analysis that were important for the sustainability classifications you list.

L145-147: 4 values of K are listed for 3 classes.

L152: ∞ symbology is incorrect here and throughout the manuscript (file error?)

L152: Should mention the general species/stocks that Musick (1999) uses so we can assess how comparable these approaches are.

L155: how were these thresholds identified?

L158-168: while this citation seems appropriate for this paper, there are several terms that could use some brief additional detail to help the reader follow. For example, the meaning of “adaptive capacity” is unclear. What types of variables fit this criteria?

L192-193: were these ecoregions sampled equally? It would be very beneficial to include a figure with a map showing the ecoregions from which data were collected.

L194-196: the database is not currently accessible via Google Drive.

L197: Table 1 does not seem necessary to include in the main body, especially since the INMARLH index in referenced sparingly in the manuscript. This table could easily be move to supplemental files. The last two rows in the table are included despite no papers referred and can be deleted. Is there a reason to describe this index in such detail given how little it is used in the manuscript?

L202: this point is worth bringing up in the discussion; does data availability here limit or bias the results in any way? What other attributes, if any, do you think could improve these calculations? You could go into greater detail on L561, for example.

L205-212: Table 2 should be in the supplemental file rather than the main body text.

L206, 213-216: Figures 1a & 1b do not seem necessary to me as presented. It would be good to indicate in the text, while describing the variables, the sample size (N) for each. You could likewise describe the distributions of scores (means, SDs, ranges, etc.) for each attribute in a table in the results. If you want to include such figures in the supplemental materials, the quality needs to be improved.

L224: This caption should do more to clarify what high scores for resilience and vulnerability indicate.

Table 3: The logic column would benefit from citations for many of the statements (these could also be included in the attribute descriptions that follow). Is the scoring system for landing price backwards (see also L309-310)?

L242: how many species required use of this proxy?

L267-268: clarify what is meant by “maximally distributed”.

L278-283: it is unclear how we should interpret these scores. How did you decide on cutoffs to adjust distributions to rankings?

L315-316: can delete this sentence.

L326: parenthetically indicate whether x-axis refers to resilience or vulnerability.

L343: “quickly” should be “accurately”.

L369-371: there needs to be more elaboration on your methodology. It is hard to follow what was done here.

L381: doesn't your IRV only represent Indian stocks? How comparable are IUCN data from other regions in these cases? That needs to be considered when comparing these indices.

Table 4: I like the idea of this table. To make it easier to follow, you might consider using vertical lines to separate the resilient, vulnerable, and risky sections.

L427-429: Figure 3 is very helpful for describing overall patterns. The separate plots (Figures 4-7) work better for interpretation; I suggest keeping Fig 3 and combining Figs 4-7 into Figs 4a-d. How are Figs 4a and 4b are currently split? Is there a reason for this? Make sure to reference Table 2 (or if it move to supplemental table) in the figure caption so readers can identify the species shown on each plot.

L434 and L453: You start to describe reasons for the classification…this should go in the discussion.

L471: should we be concerned that the most resilient species has an IRV of 0.52, and not closer to the theoretical maximum value? What aspects of the biological and fishery attributes for this species give it the highest IRV? Should we be concerned about Indian fish stocks overall? – all things to be considered in the discussion.

Figure 8: This figure could either be broken into descriptive statistics (for example, can include with L473-475) or would be improved by color coding the bars in a stacked histogram to show the distribution of IRV scores by family.

L481: “scores” = IRV scores?

L483-484: why use the 6 most sensitive attributes and not 1, 3, 5, etc? Justify the cutoff and selection process used.

L485-486: why evaluate only the 10 most resilient and vulnerable species, rather than compare IRV and sIRV for all 133 species? In Fig 9, the classification of species doesn't change for any of the species shown, but what about intermediate ones?

L500: why were these 11 species chosen for comparison?

L504-505: do differences in these indices show any trends across particular groups (elasmobranchs, etc)?

L540-541: this idea needs to be elaborated on further in the discussion.

L576: curious if there is any concern about gear bias in the results. Related to this, is gear regulation (e.g., changing mesh sizes) an option for targeted management of at-risk/vulnerable/less resilient fisheries?

L586-587: were there any temporal trends in the other variables besides Er?

L609-610: CVs (or SDs) are great for describing variation in the data; these would be better reported in the results section and interpreted here.

L616: also depends on the severity of the impact, right?

L635-636: do people use smaller mesh size when targeting smaller species? Does this minimize the effect described here?

L666-667: Evaluations and reporting of statistics for these comparisons should be made in the results section.

Figures 10 & 11ab: These should be referenced and described in the results (maybe ~L505).

L692: what is meant by “marginal”?

L698: it is unclear what is meant by “inconsistent weights will cause bias in the analysis”.

L712-713: statements such as “Many clupeids and crustaceans fall under high K category” require citations.

L737: thoughts on the possible mechanism of this relationship between resilience and geographic distribution? Is there evidence of such relationships from other studies?

L761: link to a citation or source or IUCN evaluations.

L766-767: what is meant by “evaluate its usefulness”? How was this done in the manuscript? You showed that it was possible to calculate an index, but I do not see an evaluation of usefulness.

L789-790: advantages/disadvantages of weighted and non-weighted approaches?

L801: seems that interpretation of this metric is thus relative to the fishery being studied. This has implications for comparing IRV to other indices.

L818: not following how 14 species are on the top-10 list.

L833: how often is this list revised, and could your work be used to help India reconsider which species to include on it?

L887: need to describe this “appropriate tool” in a little more detail, so readers can understand how these IRV values for multiple fisheries can be combined.

Reviewer #3: PONE-D-20-34956

General comments

The paper is a compilation of life history characteristics of fish in Indian waters that are important for fisheries and some evaluation of their potential resilience. It seems to follow on from a previous publication that looked at fewer species and metrics. Thus, the goals of the paper seem redundant with previous studies and the goals are not that clearly stated other than knowing resilience of fish is a good thing. The paper is therefore quite methodological and considerably less conceptual, hypothesis-driven and in presenting original findings. This may be fine for the journal PloSone. However, the organization of the paper is quite weak and this leads to low comprehension while reading. This is due to a mixture of things including mixing text that should go in various sections of a scientific paper. Much of the introduction is method, for example. The order is not great in the sense that many terms are not well defined and only presented as mathematical formulas late in the text. Many abbreviations and acronyms are used and not always that clear in text, tables, and figures. The English composition is also weak in many places in terms of the structure of presentation of information. The result is a text that is difficult to understand in terms of the potential importance and relevance to science and management. The paper might be better as a management report to the fisheries for information on developing fisheries policies. As a scientific paper, it is marginal. Some issue could be rectified but this will require considerable work on these and other issues.

Comments while reading

The title makes me wonder what are the attributes being evaluated as the abstract seems to be about fish less than fisheries.

Abstract

The abstract seems quite methodological and reports few results or concepts that might support some management need or scientific hypotheses. The authors say this method develops insights but we are not told what they are. I would suggest rewriting it to focus more on results and value to fisheries and climate disturbances.

L26- I am wondering what is the difference between vulnerability and risk for species? Risk assumes you know something about the frequency of disturbance. Any good reason to pick 10 species? Where does risk come in the methods and results?

L31 – the word shortened seems a poor word choice.

Introduction

Resilience or vulnerability are often specific to the disturbance. It is not that clear whether the disturbance here is fishing or climate and if this would make a differences. Can some text be used to make this clearer?

L56 – There are many more recent and thorough surveys of fish biomass that would seem appropriate to briefly review. The big criticisms of the Worm et al. 2009 paper was the lack of tropical fisheries data. So, this is a good place to argue for the originality of this paper.

P89 – it would be good to briefly say something about the spatial extent and habitats of this fisheries and boats. Is this within the EEZ or some other spatial scale? The species lists tell me this is pelagic and soft bottom species but this could be clearer to help the reader understand the fisheries environmental context.

P111 – this paragraph is not well referenced and it is not certain what habitats are being discussed here. It sounds like pelagic fisheries as many benthic fish have long lives for example.

P119 – this is an interesting finding and possibly one reason that some people are recommending gear rather than stock management. Would this be better in the discussion section?

The paper seems to be an update of species and some index of a previous study. Thus, the scientific value of the paper is limited. It seems more of an effort to increase the compilation of information of these fisheries.

Table 2 – I wonder if the information in this table would be more useful if organized by habitat or life histories or some other characteristic other than the alphabetical order of the families?

P119 – onward. Much of this text seems like methods in what is a long introduction that never is very clear about the scientific goals. I suggest a section in the methods section after the data and species description that summarizes the metrics in terms of their history and use in this paper. Otherwise, the introduction is too long and not clear.

L213- onward. The meanings of resilience, vulnerability, etc need to be defined earlier in the paper before presenting results.

Results

The results are very hard to follow because the legends are in the main text and the figures are at the end of the paper. There are also so many acronyms in the paper that this leads to poor comprehension.

The results and the redundancy of this with previous studies, suggests to me that maybe the authors should focus mostly what is original here, which I believe is the IRV index.

Table 3 comes before we know what resilience and vulnerability are and so it is unclear which of these variables belong to which metric.

Some of the text in the results seems to be more appropriate for the discussion section. That is when results are being compared to other studies.

Discussion

The discussion section is not that well organized both sub-headings and better English composition would help. That is better lead sentences on paragraphs and subsequent focused text.

6. PLOS authors have the option to publish the peer review history of their article (what does this mean?). If published, this will include your full peer review and any attached files.

Reviewer #1: No

Reviewer #2: No

Reviewer #3: No

---

## [Author Response · Author response to Decision Letter 0]

4 May 2021

Reviewer #1

1 I could not access the attached data file, so I could not fully evaluate the data. 

We did a recheck and modified the link. The difficulty may be because the file was in MS Access format (mdb file), and therefore, we have added an MS Excel format file too so that it can be easily accessible to all.

2 Very few statistics were presented. Most of the results were presented as categories of resilience or vulnerability. I suggest the authors consult biostatisticians to consider appropriate statistical tests. 

As suggested we have added descriptive statistics (new Table 1 and 6) for each of the 13 attributes used for deriving R and V scores (Line # 195/481). 

ANOVA has been done to test for differences in IRV scores between major resource groups (Table 5; Line # 470).

3 The manuscript was much longer than it needed to be and had too many small sections. It should be streamlined. Also, the authors spent too much time describing the data in the tables and figures. They would have a stronger manuscript if they spent more time interpreting their results instead of repeating what readers can see in a table or figure. 

Several edits have been made in the MS text to reduce its overall length. 

Three figures have been deleted (instead more information presented in Tables) and five figures have been combined into one. 

The total number of pages has been reduced from 55 to 54 pages. 

4 Additional statistics would help. For example, the authors describe the range of values they calculated, but it would be useful to know the values of each quartile of the data, or median, mean, and 95% confidence interval of the data. Statistical analyses could then be conducted to compare different taxa or fisheries. 

See answer to #2 above. The descriptive statistics added are Mean, CV, Quartiles (Q1 to Q4), standard deviation etc. 

ANOVA has been done to test for differences in IRV scores between major resource groups and t-test for pairwise comparisons (Table 5; Line # 470).

5 In the Discussion, the authors suggest that these types of analyses could be conducted on a regional basis, but it seems as though they have the data to do that. Why didn't they? 

While discussing the possible applications of the IRV method we did mention that regional and fleet-based scores could prove beneficial. However, we did not do it as the MS was already too long, and adding these evaluations would make it lengthier. 

6 A major comment I have is that these types of analyses have been conducted around the world, but there are very few references to indicate that the authors have read or considered other researcher's work. 

As suggested additional references have been added. (Line # 55-56; 350; 827). 

7 The Discussion section would be strengthened by a more detailed discussion of how their results could be applied to fishery management. 

As suggested additional sentences on fisheries management applications have been added.

(Line # 826-827). 

Reviewer #2

1 In general, this manuscript would benefit from a careful proofreading, as there are numerous spelling and grammatical mistakes throughout. I highlight some of these in the Specific comments below. This may help the writers clarify their meaning in some sections of the paper, which will be helpful for the overall interpretability. 

As suggested, the MS has been reedited and proofread to reduce grammatical errors. 

2 I do not agree that resilience and vulnerability are innately opposites, as is hypothesized here. As indicated in L652-653, you expected high R with low V (or vice versa) but did not always observe this result. This does not surprise me, since the two indices you calculate (R and V), while complimentary, describe attributes of each species/fishery that are not necessary inversely related (or related at all). At points throughout the manuscript, it is implied that R and V are inversely related, and these statements need to be adjusted to either justify this expectation by the authors or to clarify their expectations if my interpretation above is incorrect. 

While we agree that resilience and vulnerability are not exactly opposites as has been shown with the data in this MS, these terms have been debated in various scientific fields as either complementary or conflicting concepts. (Line # 485). We have tried to tone down such statements throughout the MS. 

3 L22: delete “was estimated”. Changed as suggested. 

4 L28: “large-sized” should be changed to “large”. Changed as suggested.

5 L30-31: delete “by the species within the same family”. Changed as suggested.

6 L33: rather that “contradictory results”, use something like “different species vulnerabilities” or “different fish management priorities”. Changed as suggested.

7 L47: “species of diverse” should be “species with diverse”. L50: delete “number of”. L89: both mentions of “is” should be changed to “are”. 

Changed as suggested.

8 L106-107: with regard to the Hilborn et al. (2020) citation, is there any information available to allow you to describe the state of the fisheries used in this study? Are they “in poor shape” as you say? 

A new citation (Sathianandan et al., 2021) has been added on the state of the fisheries in India.

(Line # 106 to 107)

9 L115: BMSY and FMSY have not been defined elsewhere in the manuscript. 

Definitions added as suggested

(Line 119 to 121) 

10 L119-127: This paragraph brings up an important point. You describe 133 species, but use data from 644 stocks. Are there multiple stocks for some of these species? If so, how were data from different stocks combined for a single species? 

Yes, there are multiple stocks for many of the species.

Fish stock is a result of management partitioning the population of a species into time-space sub‐units that have some degree of coherence biologically and/or ecologically. Therefore, the current 644 stocks belonged to 133 species. 

The range of values for attributes of a species was considered across 644 stocks. 

11 L141-143: although the original paper is cited, I believe it would be helpful to briefly describe the attributes from the siFISH analysis that were important for the sustainability classifications you list. 

The attributes used were the same for siFISH and IRV analysis, but conceptually both the analysis were different and IRV was considered as an improvement. The changes are listed in Line # 152-156.

12 L145-147: 4 values of K are listed for 3 classes. The error is corrected 

(Line # 136 to 137) 

13 L152: ∞ symbology is incorrect here and throughout the manuscript (file error?) 

Yes, the error has been corrected throughout the MS

14 L152: Should mention the general species/stocks that Musick (1999) uses so we can assess how comparable these approaches are. 

Some of the temperate species used by Musick (1999) such as Sandbar shark, Blue grenadier, Cape hake, Bay anchovy, Orange roughy etc have been added in the text as suggested.

(Line # 144 to 145)

15 L155: how were these thresholds identified? 

Thresholds were identified based on BRPs. Text added as suggested. 

(Line # 142 to 148)

16 L158-168: while this citation seems appropriate for this paper, there are several terms that could use some brief additional detail to help the reader follow. For example, the meaning of “adaptive capacity” is unclear. What types of variables fit this criteria? 

This paragraph has been deleted as suggested by Reviewer #3 to reduce the length of the introduction section. 

17 L192-193: were these ecoregions sampled equally? It would be very beneficial to include a figure with a map showing the ecoregions from which data were collected. 

As suggested Fig.1 added to depict the regions from which data records were sourced.

(Line # 181 to 186) 

18 L194-196: the database is not currently accessible via Google Drive. 

The database has been uploaded with an Excel file (besides the MS Access mdb format) for ease of access. MS Access links were rechecked.

(Line # 179)

19 L197: Table 1 does not seem necessary to include in the main body, especially since the INMARLH index in referenced sparingly in the manuscript. This table could easily be move to supplemental files. The last two rows in the table are included despite no papers referred and can be deleted. Is there a reason to describe this index in such detail given how little it is used in the manuscript? 

As suggested Table 1 has been moved to supplementary (Table S1). The INMARLH (Indian Marine Fish Life Histories) is a database and not an index. This database was created by collating biological and fishery attributes contained in the research papers and the databases mentioned in Table S1. 

The INMARLH database is the data source of this MS. 

20 L202: this point is worth bringing up in the discussion; does data availability here limit or bias the results in any way? What other attributes, if any, do you think could improve these calculations? You could go into greater detail on L561, for example. 

We are not advocating the addition of attributes for IRV analysis, as the process would make it more data demanding and would prove restrictive. This is the reason why an abridged version of IRV with a lesser number of attributes was tested. As suggested we have added a sentence to the Discussion section on the living nature of the INMARLH database. 

(Line # 573 to 575)

21 L205-212: Table 2 should be in the supplemental file rather than the main body text. 

As suggested, Table 2 moved to the Supplemental section as Table S2. 

22 L206, 213-216: Figures 1a & 1b do not seem necessary to me as presented. It would be good to indicate in the text, while describing the variables, the sample size (N) for each. You could likewise describe the distributions of scores (means, SDs, ranges, etc.) for each attribute in a table in the results. If you want to include such figures in the supplemental materials, the quality needs to be improved. 

As suggested Figures, 1a and 1b have been deleted. Instead, the descriptive statistic data is presented as a Table (Table 1) which has the mean, CV, Std Deviation and quartiles. 

(Line # 195 to 201)

23 L224: This caption should do more to clarify what high scores for resilience and vulnerability indicate. Modified as suggested. (Line # 195 to 201)

24 Table 3: The logic column would benefit from citations for many of the statements (these could also be included in the attribute descriptions that follow). Is the scoring system for landing price backwards (see also L309-310)? 

Citation added in the Table header as suggested. The citations have been also added to the attribute descriptions. (Line # 195 to 201; 219, 226, 236 etc). Yes, the scoring for landing price is backwards – higher value fishes are targeted and hence more vulnerable. 

25 L242: how many species required use of this proxy? 

42 species – added in parenthesis. (Line # 229)

26 L267-268: clarify what is meant by “maximally distributed”. 

Phrase changed for clarity. (Line # 255 to 256)

27 L278-283: it is unclear how we should interpret these scores. How did you decide on cutoffs to adjust distributions to rankings? 

This paragraph has been modified to bring clarity. The maximum value (100) was broadly divided into 3 equal parts for distribution into 3 ranks. (Line # 268 to 272)

28 L315-316: can delete this sentence. Sentences deleted as suggested.

29 L326: parenthetically indicate whether x-axis refers to resilience or vulnerability. 

Added resilience in parenthesis for the x-axis. (Line # 314)

30 L343: “quickly” should be “accurately”. Changed as suggested. (Line # 335)

31 L369-371: there needs to be more elaboration on your methodology. It is hard to follow what was done here. 

There was an error in the comparison of FishBase vulnerability categorization as depicted in Table 10 (Line # 543 to 545). This has been corrected and sentences reframed for clarity. (Line # 359 to 366).

32 L381: doesn't your IRV only represent Indian stocks? How comparable are IUCN data from other regions in these cases? That needs to be considered when comparing these indices. 

The IUCN database is a global one and does include species that are found in Indian waters (60 among the 133 species in this study). This has been explained in the results section (Line # 546-553). 

33 Table 4: I like the idea of this table. To make it easier to follow, you might consider using vertical lines to separate the resilient, vulnerable, and risky sections. 

As suggested vertical lines have been added to Table 3 separating resilient, vulnerable and risky sections.

34 L427-429: Figure 3 is very helpful for describing overall patterns. The separate plots (Figures 4-7) work better for interpretation; I suggest keeping Fig 3 and combining Figs 4-7 into Figs 4a-d. How are Figs 4a and 4b are currently split? Is there a reason for this? Make sure to reference Table 2 (or if it move to supplemental table) in the figure caption so readers can identify the species shown on each plot. 

As suggested Figs 4a, 4b, 5, 6 and 7 have been combined as Fig.4 (a to e). Figs 4a and 4b were split almost equally among the 96 teleosts. 

35 L434 and L453: You start to describe reasons for the classification…this should go in the discussion. 

As suggested, these phrases have been deleted from the results section. 

36 L471: should we be concerned that the most resilient species has an IRV of 0.52, and not closer to the theoretical maximum value? What aspects of the biological and fishery attributes for this species give it the highest IRV? Should we be concerned about Indian fish stocks overall? – all things to be considered in the discussion. 

As per our IRV methodology, the theoretical maximum score for IRV is 1, and this is obtained when the points are close to the origin in the scatter plot (resilience 3 and vulnerability 1). However, none of the 133 species assessed by us had low vulnerability (<1.5) as shown in the contingency matrix Table 4. This point has been discussed (Line # 686-691). 

37 Figure 8: This figure could either be broken into descriptive statistics (for example, can include with L473-475) or would be improved by color coding the bars in a stacked histogram to show the distribution of IRV scores by family. 

This figure is now changed to Table 6 showing descriptive statistics. It was not possible to do a stacked bar with clarity as there are 55 families. 

38 L481: “scores” = IRV scores? Yes. Changed as suggested

(Line # 497)

39 L483-484: why use the 6 most sensitive attributes and not 1, 3, 5, etc? Justify the cutoff and selection process used. 

The concept was to have an abridged version of the IRV which could be used in data-limited situations. Since the original IRV had 13 attributes, we considered about half of the attributes as a data-limited threshold. 

40 L485-486: why evaluate only the 10 most resilient and vulnerable species, rather than compare IRV and sIRV for all 133 species? In Fig 9, the classification of species doesn't change for any of the species shown, but what about intermediate ones? 

We chose these 10 species as an indicator to test the efficacy of the abridged version of the IRV. Results of testing all 133 species would be very unwieldy to show as a Table or Figure. 

41 L500: why were these 11 species chosen for comparison? 

This decision was based on the data availability for PSA and also ensuring all 4 groups (teleosts, elasmobranchs, crustaceans and molluscs) were nominally covered. 

42 L504-505: do differences in these indices show any trends across particular groups (elasmobranchs, etc)? 

Yes, particularly for elasmobranchs. This is added now in the text (Line # 519-521).

43 L540-541: this idea needs to be elaborated on further in the discussion. 

Changed as suggested. (Line # 786-793).

44 L576: curious if there is any concern about gear bias in the results. Related to this, is gear regulation (e.g., changing mesh sizes) an option for targeted management of at-risk/vulnerable/less resilient fisheries? 

Certainly sampling from different gears over a long period is expected to introduce bias in the estimates of attributes. However, we could not discern this as stated in the discussion. There are mesh regulations for gears as a technical measure and also certain banned gears to protect ETP species. 

45 L586-587: were there any temporal trends in the other variables besides Er? 

We were not able to discern any pattern or trends. 

46 L609-610: CVs (or SDs) are great for describing variation in the data; these would be better reported in the results section and interpreted here. 

As suggested, these statements have been moved to the Results section. 

47 L616: also depends on the severity of the impact, right? 

Yes, the sentence changed as suggested. (Line # 625-626).

48 L635-636: do people use smaller mesh size when targeting smaller species? Does this minimize the effect described here? 

Yes, fishers generally use smaller mesh size than that legally prescribed. In a multispecies assemblage, this results in growth overfishing. 

49 L666-667: Evaluations and reporting of statistics for these comparisons should be made in the results section. Changed as suggested. (Line # 472-474).

50 Figures 10 & 11ab: These should be referenced and described in the results (maybe ~L505). Changed as suggested. (Line # 475-480).

51 L692: what is meant by “marginal”? 

The term has been changed to negligible for clarity. (Line # 699).

52 L698: it is unclear what is meant by “inconsistent weights will cause bias in the analysis”. 

As suggested, the sentence has been reframed for clarity. (Line # 704-705).

53 L712-713: statements such as “Many clupeids and crustaceans fall under high K category” require citations. Reference added as suggested. (Line # 720).

54 L737: thoughts on the possible mechanism of this relationship between resilience and geographic distribution? Is there evidence of such relationships from other studies? 

Line # 734-746 discusses the relationship between resilience and geographic distribution. A new reference on coral reef spatial resilience has been added. 

55 L761: link to a citation or source or IUCN evaluations. Citation added. (Line # 771).

56 L766-767: what is meant by “evaluate its usefulness”? How was this done in the manuscript? You showed that it was possible to calculate an index, but I do not see an evaluation of usefulness. Yes, the statement is wrongly applied and has been deleted. 

57 L789-790: advantages/disadvantages of weighted and non-weighted approaches? 

While weighting can give more realistic and accurate estimates, its application on a wider scale with regional differences becomes a problem. This is particularly so in data-limited situations. (not answered in MS text).

58 L801: seems that interpretation of this metric is thus relative to the fishery being studied. This has implications for comparing IRV to other indices. 

Yes, the metric is changed because of the differences in the species life-history patterns in the tropics as compared to other regions. (not answered in MS text).

59 L818: not following how 14 species are on the top-10 list. 

There are common species in the top-ten vulnerable and high-risk species (2 lists). (Line # 831-832).

60 L833: how often is this list revised, and could your work be used to help India reconsider which species to include on it? 

While the outputs of this work can be used to modify the Indian Wildlife Protection Act, there is no standard protocol; for periodic revisions of the list. 

61 L887: need to describe this “appropriate tool” in a little more detail, so readers can understand how these IRV values for multiple fisheries can be combined. 

As suggested a sentence has been added to bring more clarity on the approach adopted by Micheli et al (2014). (Line # 903-907).

Reviewer #3

1 The title makes me wonder what are the attributes being evaluated as the abstract seems to be about fish less than fisheries. 

The MS is about the resilience and vulnerability of tropical fish and not about fisheries per se. For this, the study uses, apart from biological attributes, certain fishery attributes such as exploitation rate to gauge the response of fish to a fishery. The findings do have application in fisheries management and stock assessment. 

2 The abstract seems quite methodological and reports few results or concepts that might support some management need or scientific hypotheses. The authors say this method develops insights but we are not told what they are. I would suggest rewriting it to focus more on results and value to fisheries and climate disturbances. 

The abstract has been modified with the addition of more results. The possible application of IRV to fisheries management has been edited as suggested. 

3 L26- I am wondering what is the difference between vulnerability and risk for species? Risk assumes you know something about the frequency of disturbance. Any good reason to pick 10 species? Where does risk come in the methods and results? 

This error has been corrected in the MS. We have used the term high-risk to mean those fishes which have low IRV values which are based on estimated resilience and vulnerability. 

Top-10 is common usage in listings to grab attention and encourage debate. 

Modifications have been made in the methods and results to indicate high-risk. (Line # 316-318; 400-401).

4 L31 – the word shortened seems a poor word choice. 

As suggested the shortened terminology has been changed to abridged IRV (aIRV)

5 Resilience or vulnerability are often specific to the disturbance. It is not that clear whether the disturbance here is fishing or climate and if this would make a differences. Can some text be used to make this clearer? 

All statements in the introduction and elsewhere in the MS pertain to impacts of fishing and not to climate. To bring clarity we have added definitions of resilience and vulnerability in the introduction. 

(Line # 156-159).

6 L56 – There are many more recent and thorough surveys of fish biomass that would seem appropriate to briefly review. The big criticisms of the Worm et al. 2009 paper was the lack of tropical fisheries data. So, this is a good place to argue for the originality of this paper. 

As suggested, recent references (more than 5) have been added throughout the MS to bring out the importance of this MS from a data-deficient and tropical fisheries setting. 

7 P89 – it would be good to briefly say something about the spatial extent and habitats of this fisheries and boats. Is this within the EEZ or some other spatial scale? The species lists tell me this is pelagic and soft bottom species but this could be clearer to help the reader understand the fisheries environmental context. 

Additional information on the EEZ has been added (Line # 93, 182). Besides a map (Fig.1) has been provided to indicate the spatial distribution of the data records as requested by Reviewer #2 (#17).

Table 2 which was the species listing (now moved to supplementary information, Table S2, as per Reviewer #2) has an extra column on habitat.

8 P111 – this paragraph is not well referenced and it is not certain what habitats are being discussed here. It sounds like pelagic fisheries as many benthic fish have long lives for example. 

As suggested the species groups are now added (Line # 114-115). Mainly large benthic fish are long-lived in the tropics. 

9 P119 – this is an interesting finding and possibly one reason that some people are recommending gear rather than stock management. Would this be better in the discussion section?

This paragraph has been deleted from the introduction section and summarized in the discussion section. 

10 The paper seems to be an update of species and some index of a previous study. Thus, the scientific value of the paper is limited. It seems more of an effort to increase the compilation of information of these fisheries. 

The earlier work was rudimentary with a limited period of data and conceptually different. The present analysis attempts to provide a comparative index for tropical fish species. 

11 Table 2 – I wonder if the information in this table would be more useful if organized by habitat or life histories or some other characteristic other than the alphabetical order of the families? 

The listing in Table 2 (now moved to supplementary, Table S2) is primarily classified into 6 groups and then according to the alphabetical order of families. The resulting R-V plots are split according to the primary grouping for ease of understanding. The numbers in the plots refer to the species number in this table. Habitat information on each species has been added to Table S2.

12 P119 – onward. Much of this text seems like methods in what is a long introduction that never is very clear about the scientific goals. I suggest a section in the methods section after the data and species description that summarizes the metrics in terms of their history and use in this paper. Otherwise, the introduction is too long and not clear. As suggested, 2 paragraphs describing the earlier work have been deleted. The earlier work is referenced (accessible to readers) and therefore it is not repeated. 

13 L213- onward. The meanings of resilience, vulnerability, etc need to be defined earlier in the paper before presenting results. 

Resilience and vulnerability as used in this MS are now defined. (Line # 156-159). 

14 The results are very hard to follow because the legends are in the main text and the figures are at the end of the paper. There are also so many acronyms in the paper that this leads to poor comprehension. 

Sorry about this!! But this is the way the journal organizes the MS. All acronyms are given in full the first time it is used. 

15 The results and the redundancy of this with previous studies, suggests to me that maybe the authors should focus mostly what is original here, which I believe is the IRV index. 

Agreed. We have edited the MS with the suggested focus. 

16 Table 3 comes before we know what resilience and vulnerability are and so it is unclear which of these variables belong to which metric. 

Resilience and vulnerability are now defined in the Introduction section.

17 Some of the text in the results seems to be more appropriate for the discussion section. That is when results are being compared to other studies. 

Specific instances have been remarked by Reviewer #2 and changes have been made as suggested. 

18 The discussion section is not that well organized both sub-headings and better English composition would help. That is better lead sentences on paragraphs and subsequent focused text. 

By revising the manuscript following the suggestions of the 3 Reviewers, we believe that the presentation of the manuscript including Discussion has improved substantially.

---

## [Decision Letter · Decision Letter 1]

27 Jul 2021

Application of biological and fisheries attributes to assess the vulnerability and resilience of tropical marine fish species

PONE-D-20-34956R1

Dear Dr. Sathianandan,

We’re pleased to inform you that your manuscript has been judged scientifically suitable for publication and will be formally accepted for publication once it meets all outstanding technical requirements.

Kind regards,

Even Moland

Academic Editor

PLOS ONE

Additional Editor Comments (optional):

I encourage the authors to replace the two key words which are present in the title of the manuscript (resilience and vulnerability) and thus redundant as key words. Replacing these with other relevant key words will ensure increased searchability of the published paper. Suitable replacement key words for consideration could be e.g., 'Ecosystem Approach to Fisheries Management (EAFM)' and 'fisheries dependent data'.

---

## [Editor Report · Acceptance letter]

5 Aug 2021

PONE-D-20-34956R1 

Application of biological and fisheries attributes to assess the vulnerability and resilience of tropical marine fish species 

Dear Dr. Sathianandan:

I'm pleased to inform you that your manuscript has been deemed suitable for publication in PLOS ONE. Congratulations! Your manuscript is now with our production department. 

Kind regards, 

on behalf of

Dr. Even Moland 

Academic Editor

PLOS ONE